# Scaling Vision-and-Language Navigation With Offline RL

**Valay Bundele**[†]    *valaybundele@gmail.com*
*University of Tübingen*

**Mahesh Bhupati**    *mail2mahesh0537@gmail.com*
*Indian Institute of Technology Bombay*

**Biplab Banerjee**    *bbanerjee@iitb.ac.in*
*Indian Institute of Technology Bombay*

**Aditya Grover**    *adityag@cs.ucla.edu*
*University of California, Los Angeles*

**Reviewed on OpenReview:** *https://openreview.net/forum?id=kPIU8PnJPo*

## Abstract

The study of vision-and-language navigation (VLN) has typically relied on expert trajectories, which may not always be available in real-world situations due to the significant effort required to collect them. On the other hand, existing approaches to training VLN agents that go beyond available expert data involve data augmentations or online exploration which can be tedious and risky. In contrast, it is easy to access large repositories of suboptimal offline trajectories. Inspired by research in offline reinforcement learning (ORL), we introduce a new problem setup of VLN-ORL which studies VLN using suboptimal demonstration data. We introduce a simple and effective reward-conditioned approach that can account for dataset suboptimality for training VLN agents, as well as benchmarks to evaluate progress and promote research in this area. We empirically study various noise models for characterizing dataset suboptimality among other unique challenges in VLN-ORL and instantiate it for the VLN↻BERT and MTVM architectures in the R2R and RxR environments. Our experiments demonstrate that the proposed reward-conditioned approach leads to significant performance improvements, even in complex and intricate environments.[1]

## 1 Introduction

Vision-and-language navigation (VLN) is a complex task that requires an agent to navigate through environments based on language instructions, involving visual perception, natural language processing, and sequential decision making. VLN has received considerable attention (Fu et al., 2020b; Wang et al., 2019; Ma et al., 2019; Fried et al., 2018) in recent years due to its potential applications in robotics, autonomous vehicles, and virtual assistants. However, this task poses significant challenges, such as dealing with ambiguity in language instructions, robust perception in dynamic environments, and efficient exploration in large-scale environments.

The key challenge in VLN is learning a generalizable policy given diverse vision and language inputs, so scaling training data is crucial for the performance of VLN agents. However, collecting and annotating VLN data is difficult, leading to a shortage of domain-specific training data that limits performance in unseen environments. To address this challenge, some works have performed data augmentation by synthesizing new instructions (Fried et al., 2018) or varying appearance of existing environments (Li et al., 2022). Further, many works use pre-trained models trained on large language and vision datasets using self-supervised

---

[†]Work done when the author was at Indian Institute of Technology Bombay.
[1]Code and datasets available at `https://github.com/Valaybundele/RewardC-VLN-ORL`

objectives (Hao et al., 2020; Hong et al., 2021; Guhur et al., 2021; Zhu et al., 2020). Some works have also acquired extra data through online exploration (Hong et al., 2021; Lin et al., 2022a; Chen et al., 2021b). However, exploration can be dangerous in many real-world applications such as robots in safety-critical applications or healthcare problems. Specifically, in VLN, exploration can be dangerous if the agent misunderstands instructions or distribution shifts in visual inputs, leading to collisions with walls or humans. Given these limitations, our work centers on a fundamental question: *How can we achieve effective data scaling without resorting to online exploration, a strategy that could pose safety concerns?*

To this end, this work makes the observation that while expert data can be hard to acquire, we often have access to large databases of subotimal offline trajectories. Examples include: 1) Human Navigation Data: In congested urban settings, drivers often take suboptimal routes due to traffic, road closures, or parking constraints, providing a rich source of suboptimal navigation data. 2) Imperfect Simulated Environments: AI agents navigating simulated environments, like a shopping mall, may encounter dynamic obstacles and adjust their paths accordingly. These adaptations, driven by imperfect navigation algorithms, could result in suboptimal trajectories. 3) Transfer Learning Scenarios: Consider a robot model trained in a laboratory setting and then deployed in a real-world hospital. Initially, the robot might follow suboptimal paths as it learns to navigate the unique challenges posed by the hospital's layout and unexpected obstacles before fine-tuning its navigation strategy. These are just a few illustrative examples, but in reality, a multitude of scenarios can provide suboptimal demonstration data with relative ease. This accessibility to suboptimal data presents a crucial opportunity for advancing VLN research, surpassing the challenges associated with obtaining expert data and propelling progress in the field.

In recent years, learning from logged demonstration data has become an active area of research within the offline RL community. While many algorithms exist including value-based (Peng et al., 2019; Wu et al., 2019; Siegel et al., 2020; Kumar et al., 2020), model-based (Kidambi et al., 2020; Janner et al., 2019), and model-free (Chen et al., 2021a; Emmons et al., 2021) approaches for training agents from logged data, there has been limited work in extending these methods to challenging VLN domains. *In this paper, we propose the setting of vision-language navigation with offline RL (VLN-ORL) which trains VLN agents efficiently using offline datasets.* Given that most current offline VLN algorithms rely on supervised learning from expert datasets, the key practical goal for VLN-ORL is to design algorithms that seamlessly align with the established VLN architectures and objectives in use today. Therefore, we propose to extend family of approaches based on RL via supervised learning (RvS) (Emmons et al., 2021).

RvS approaches learn an agent policy from a sub-optimal dataset by conditioning the policy on a outcome variable, e.g., returns-to-go or the goal token. At test time, the agent is conditioned on the desired outcome, e.g., expert returns. For VLN, we do not expect to know the goal coordinates at test-time and need to resolve the language instruction to a goal state. Prior work resolves this challenge by learning to map the agent state to a reward function that can be maximized by the agent using an off-the-shelf RL algorithm (Nair et al., 2022). However, this assumes the existence of a sufficient number of expert trajectories in the offline dataset, which might not be true. In contrast, we propose an alternate set of assumptions that require knowledge of the goal state only during training, even if it is not achieved by the agent in executing an instruction. We build on this assumption to design a novel *reward token* which allows flexible conditioning of VLN agents during training and testing.

Our next contribution is to establish the first set of offline RL benchmarks for evaluating VLN algorithms. Specifically, we leverage rollouts of a HAMT policy (Chen et al., 2021b) pre-trained on R2R (Anderson et al., 2018) and RxR (Ku et al., 2020) datasets and develop several noise models to perturb the agent rollouts and generate suboptimal datasets of varying difficulties. Furthermore, we employ the proposed *reward token* to condition two recent VLN agents, namely VLN↺BERT and MTVM. Through the utilization of reward-conditioning, we observe a significant performance improvement, with an average increase of approximately 10-15% across most of the generated datasets. Notably, on the Random-policy R2R dataset, which features trajectories generated by a random policy, the performance of VLN↺BERT exhibits a remarkable improvement of approximately 40%. In addition to these results, we evaluate our approach separately on subsets of varying difficulties of the RxR validation sets, revealing that our reward-conditioning technique consistently leads to performance improvements, even in complex and intricate environments. Moreover, our findings

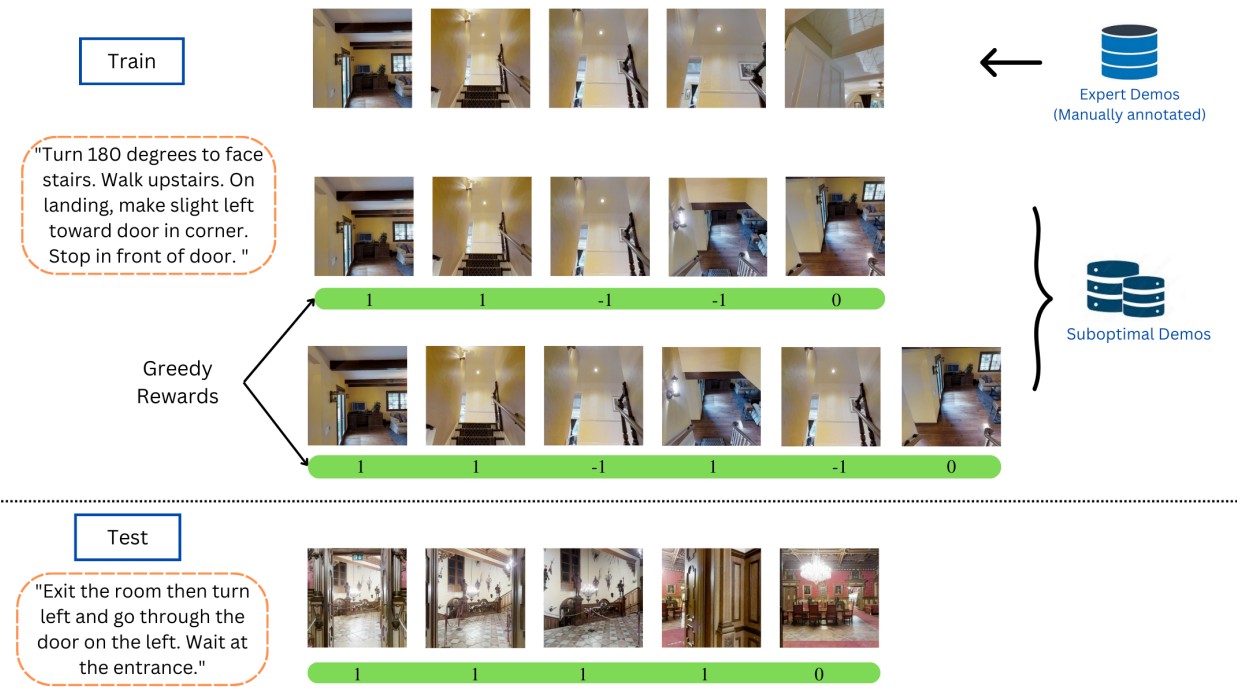

Figure 1: Illustration of proposed setup and algorithmic framework for VLN-ORL. The proposed setup involves training the agent on a dataset primarily comprised of suboptimal demonstrations of varying lengths (middle two rows). Unlike conventional VLN setups that rely solely on expert trajectories (top row), we condition our agent on greedy rewards during training to capture the degree of suboptimality without imposing excessive assumptions on the environment. Positive rewards guide the agent towards the goal, negative rewards steer it away, and zero rewards prompt the agent to halt its movement. During testing, we can condition the agent on optimal greedy rewards for executing new instructions successfully (last row).

indicate that reward-conditioning fosters the development of a more robust model, one that is less sensitive to variations in the training data.

To summarize, our **major contributions** are as follows:

- We introduce a novel problem setup, VLN-ORL, aimed at scaling VLN agents through the utilization of extensive suboptimal demonstration data.

- We design various noise models for generating suboptimal datasets for R2R (Anderson et al., 2018) and RxR (Ku et al., 2020) environments.

- We propose a simple and effective approach for extending any existing VLN agent architecture to train on suboptimal demonstration data via reward-conditioning.

- We empirically demonstrate that reward-conditioning can significantly improve the performance of VLN agents trained on suboptimal demonstrations, even in complex and intricate environments.

Since we propose using suboptimal datasets for effective data scaling, there may be concerns about the potential introduction of unsafe actions. However, it's important to note that while imperfect demonstrations may not follow optimal routes, not all of them lead to unsafe actions. The offline RL setting assumes the existence of a logged dataset of imperfect demonstrations. Notably, this dataset is not collected by the

learning agent itself; instead, it is assumed to be collected by other agents or behavioral policies. Hence, the agent we are trying to learn is not taking unsafe actions, since it is not even collecting the offline data. More generally, our main contribution in this work is not related to questioning how these imperfect demonstrations would be collected; it is more related to the question of how one can learn a policy from suboptimal offline datasets without any further exploration. This aligns with the common setup in prior works on offline RL (Fu et al., 2020a; Fujimoto et al., 2019), where the emphasis is placed on learning from historical data without requiring additional exploration during the training phase.

## 2   Related Works

**Vision-language navigation:** In recent years, there has been significant interest in developing visually grounded agents that can follow language instructions. One notable contribution is the introduction of R2R dataset (Anderson et al., 2018), where a sequence-to-sequence LSTM model was used to solve the VLN task. A critical challenge in VLN is to develop agents that can generalize well in unseen environments. Since training data is limited, many approaches use data augmentation or introduce auxiliary tasks for improving data efficiency. Speaker-follower (Fried et al., 2018) models synthesize new instructions for data augmentation and implement pragmatic reasoning. EnvDrop (Tan et al., 2019) proposes an environmental dropout strategy to generate new environments. AuxRN (Zhu et al., 2020) trains the agent with self-supervised auxiliary reasoning tasks to leverage rich semantic information in the dataset. EnvEdit (Li et al., 2022) generates new environments by varying style, object appearance and object classes of existing environments. *Our work is complementary to these works and can also benefit from augmentation strategies.*

In a related direction there has been an increase in the use of pre-trained vision-language models (Chen et al., 2020; Li et al., 2020; Su et al., 2019; Li et al., 2019) for tasks related to VLN (Hao et al., 2020; Hong et al., 2021). PREVALENT (Hao et al., 2020) pre-trains a vision-language encoder on a large number of image-text-action triplets in a self-supervised manner and fine-tunes it on downstream tasks. VLN↻BERT (Hong et al., 2021) implements a recurrent function to store navigation history in state token. Follow-up works include ADAPT (Lin et al., 2022a) which provides the agent with action prompts to enable explicit learning of action-level modality alignment. HAMT (Chen et al., 2021b) uses long-horizon history encoded using a hierarchical vision transformer along with text and current observation to predict next action. AirBERT (Guhur et al., 2021) gathers a vast amount of image-caption pairs to train for path-instruction matching task. *These methods perform online exploration in addition to learning from manually annotated R2R and RxR datasets to train the agent effectively. However, we focus on developing techniques that can learn from offline sub-optimal datasets reducing the need for exploration and developing manually annotated datasets.*

**Offline RL**: Offline RL is a subfield of RL where the agent learns a policy from a fixed dataset of transitions without any interaction with the environment. Imitation-style approaches like behavioural cloning (BC), filtered BC, and weighted BC learn a policy that mimics actions of the offline dataset. A recent class of approaches (Chen et al., 2021a; Kumar et al., 2019b) are based on conditional BC which involves conditioning the learned policy on a context variable such as the reward, the goal state or the expected return. We focus on these methods here for their simplicity, effectiveness, and direct use in our paper, while deferring the survey of additional works in value and model-based methods for offline RL to Appendix C. Kumar et al. (2019b) directly condition the agent's policy on external reward signals to optimize its behavior in RL tasks, demonstrating the potential of using any collected experience as optimal supervision when conditioned on policy quality. Decision Transformer (Chen et al., 2021a; Zheng et al., 2022) parameterizes the conditional BC policy via a transformer conditioned on rewards-to-go token. RvS (Emmons et al., 2021) shows that the performance of such methods greatly depends on model capacity and conditioning information. Recent works (Janner et al., 2022; Wang et al., 2022; Ajay et al., 2022) have combined diffusion models with RL algorithms to solve sequential decision-making tasks. *Our work sets itself apart from these conditioned BC approaches by introducing a novel reward token specifically tailored for flexible conditioning of VLN agents.*

Several works have previously introduced tasks and datasets to benchmark progress in offline RL. They have mainly used pre-trained behavioural policies ranging from a random initial policy to a near-optimal policy for generating datasets. D4RL (Fu et al., 2020a) introduces a range of dataset collection procedures based on the replay experiences and rollouts of unconverted pre-trained policies for physical environments such as

MuJoCo. Singh et al. (2022) argue that practical applications of RL involve learning from datasets with high behaviour variability across the state space. Zhou et al. (2022) use a hand-designed expert policy to collect data for real-world robotics tasks. ReViND (Shah et al., 2022) introduces an offline RL system for long-horizon goal-conditioned robotic navigation where the goal is a 2D GPS coordinate expressed in the agent's frame of reference. Nair et al. (2022) learns vision-based language-conditioned manipulation tasks from entirely offline datasets. While acknowledging the impact of these works, *our work differs from them as we introduce the first offline RL system for vision-language navigation in 3D environments.*

## 3 Preliminaries

**Setup:** We consider a multi-task extension of the Markov Decision Process (MDP), which can be represented as $M = \{S, A, P, R, p(T), H\}$. Here, $S$ represents the set of states, $A$ refers to the discrete set of actions, $P$ denotes transition dynamics, $R$ represents the task-conditioned reward function which can depend on one or more states, $p(T)$ denotes a distribution over tasks specified by language instructions and $H$ refers to the episode horizon. The term "episode horizon" refers to the predefined maximum duration or length of an episode during which an agent interacts with its environment to achieve a specific goal. Let $I$ denote the set of language instructions and $I_m$ refer to the subset of instructions corresponding to a task $T_m$. We have access to an offline dataset of $N$ trajectories, where trajectory $\tau_n = [((s_{0n}, a_{0n}), (s_{1n}, a_{1n}), ..., (s_{fn}, a_{fn})), i_n]$ consists of a sequence of state-action pairs along with the instruction $i_n$, having final state $s_{fn}$ for some positive integral $n <= N$. The state refers to the tuple, $s_t = \{C_t, \theta_t, \phi_t\}$, where $C_t$ represents the 3D position and $\theta_t$ and $\phi_t$ represent the angles of heading and elevation respectively. Given the language instruction $i_n$, the agent observes a panoramic view of the environment at every timestep $t$ and takes an action $a_t$ which takes it from state $s_t$ to state $s_{t+1}$. Specifically, the agent chooses one of the navigable locations returned by the environment at every timestep. The agent executes actions sequentially to navigate the environment and takes the stop action when it reaches the goal location.

**Environment:** We primarily work with the Matterport3D environment Chang et al. (2017) in this work. The Matterport3D simulator is a large-scale interactive RL environment constructed from the Matterport3D dataset. It consists of $10,800$ panoramic views constructed from $194,400$ RGB-D images of 90 building-scale scenes. The Room-to-Room (R2R) dataset contains $21,567$ open-vocabulary, crowd-sourced navigation instructions collected from Matterport3D. Each instruction provides a route that spans multiple rooms and consists of about 29 words on average. For instance, an instruction from the R2R dataset might read: "*Proceed upstairs, pass the piano through the archway straight ahead, turn right at the end of the hallway where the pictures and table are located, and stop at the moose antlers on the wall.*" Successfully following such instructions requires a good language understanding of spatial concepts in the real world and the ability to interpret 3D visualizations.

The RxR dataset is also built upon the Matterport3D simulator. It comprises $33,954$ English instructions with an average length of around 100 words. When comparing paths in RxR and R2R, we note that RxR paths tend to be longer, spanning an average of 8 edges and 14.9 meters, whereas R2R paths span an average of 5 edges and 9.4 meters. Additionally, variation in length between different RxR paths is much greater than that of R2R paths.

## 4 Methodology

Since VLN tasks involve navigating through large, complex environments, it can be difficult to collect sufficient amounts of high-quality data through online interactions alone. With offline RL, however, agents can learn from diverse experiences that may not be feasible to collect through online interactions. In this regard, we propose a new setup called VLN-ORL, where VLN agents are trained on suboptimal logged demonstrations. Precisely, there are 2 key challenges that VLN-ORL seeks to combat: (a) choice of offline RL algorithm, and (b) design of suitable reward token for conditioning.

**Choice of offline RL algorithm:** Value-based (Peng et al., 2019; Wu et al., 2019; Siegel et al., 2020; Kumar et al., 2020) and model-based approaches (Kidambi et al., 2020; Janner et al., 2019) to offline RL are either unstable to train or suffer from compounding errors during planning (Chen et al., 2021a). Incorporating

such approaches in available VLN techniques to work with offline datasets would be a cumbersome task. In contrast, conditioned BC approaches have shown performance at par with value-based approaches and are easier to scale with as they are trained using architectures and objectives similar to supervised learning (Reed et al., 2022; Lee et al., 2022). The key idea here is to condition the policy on a suitable variable, e.g., return or goal state. At test time, we set the conditioning to the desired value of the variable e.g., an expert return value. In this work, we will focus on extending conditional BC approaches for VLN-ORL through design of a practical reward token.

**Designing reward tokens for conditional VLN agents:** The key challenge for VLN-ORL is to characterize the degree of suboptimality in offline trajectories. To this end, we need to define a reward token that can be computed easily during training and used to specify desired optimal behavior during testing. This requires assumptions on the observability structure of the environment. Guided by practical concerns, we propose a minimal set of 3 assumptions. First, we note that general language instructions can be ambiguous (e.g., "go to the room" can refer to any room in a big house). To avoid unidentifiability in reward design, we assume that the language instruction for the training trajectories corresponds to a single pair of 2D goal coordinates. Second, we assume that during training, we have access to the agent's current coordinates and goal coordinates. Third, we assume that at test time, the agent can detect completion of a language instruction, e.g., the agent might observe a red sign at the goal location to indicate success.

We now make a few remarks to justify and motivate these assumptions. Note that the first two assumptions apply only for the training trajectories in our offline dataset. At test time, we allow the mapping between language instructions and goal states to be one-to-many, and do not assume any knowledge of the goal coordinates. During training, these assumptions are necessary to define a suitable reward function (discussed below). The third assumption allows the agent to terminate the episode and avoid looping behavior, e.g., cycling through one or more goal room(s).

Given the above set of assumptions, we can now define a reward token for conditioning the agent during training and testing. The reward token relates to the presumed reward at each state, which can be either sparse or dense. Typically, sparse rewards for VLN indicate whether a task was accomplished (Nair et al., 2022). In offline RL, the majority of demonstrations are typically suboptimal and hence, relying on a sparse reward signal for conditioning the agent may not generalize well at test-time. However, if we have access to geometric map of the environment and coordinates of the agent state and the goal state, one can compute a dense reward function simply based on shortest path distances and optimize it via a path planning algorithm. However, we do not assume the map is known, as is typical for complex real-world environments, especially for multi-task open-world language guided scenarios at test-time.

Previous research (Ng et al., 1999) has shown that potential-based rewards, derived from a potential function defined over the environment's states, can ensure the learned policy remains optimal relative to the original reward function. These rewards are based on the difference in potential function values between states, providing a way to shape the agent's behavior without altering its optimal policy. Leveraging the principles of potential-based rewards, we propose to bridge the gap between sparse and dense rewards by conditioning the agent policy on a suitably shaped reward token, referring to *change in displacement* across two consecutive states, denoted as $\delta D$. Let $D(s, G)$ represent the Euclidean distance from an arbitrary state $s$ to the goal state $G$. We define a dense version of reward token $\delta D$ for use in training as:

$$\delta D_{t,\text{ train}}^{\text{dense}}(s_t, s_{t+1}, G) = D(s_t, G) - D(s_{t+1}, G)$$

for $t \geq 0$. A positive value indicates that the agent gets closer to the goal whereas a negative value indicates that the agent is heading further away from the goal at time $t + 1$. A value of 0 indicates no displacement, which would be the reward when the agent has reached the goal. We make 2 remarks here: First, note that $\delta D_{\text{train}}^{\text{dense}}$ is a greedy measure of success, as it is possible that in some cases, the shortest path for an agent might involve navigating to locations that are further from the goal but open pathways that are significantly shorter in the long run. However, the positive side of this greedy approximation is that we do not require knowing full map of the environment during training. Second, evaluating $\delta D_{\text{train}}^{\text{dense}}$ requires knowledge of the next state $s_{t+1}$. While this is available during training in our offline trajectories, we cannot possibly know its value at test-time without executing an action at time $t$. We can handle this issue at test-time by identically

conditioning the model on a fixed positive value, without loss of generality, say $+1$, and 0 when the agent has reached the goal.

$$\delta D_{\text{t, test}}(s_t, s_{t+1}, G) = \begin{cases} 0 & \text{if } s_t = G \\ 1 & \text{otherwise} \end{cases}$$

The justification for this test-time conditioning is that at test time, we desire the agent to execute actions that move it closer to the goal. Again, there can be false positives for this greedy conditioning but it allows us to make lesser assumptions about knowing the environment map and does not require having a predictor for next states. One issue that can occur in practice is that $\delta D_{\text{train}}^{\text{dense}}$ and $\delta D_{\text{test}}$ may not be on the same scale across different instructions and environments. This can lead to a distribution shift. To mitigate this issue, we propose a sparse scale-agnostic variant of the reward token that simply looks at the sign of the change in displacement:

$$\delta D_{\text{t, train}}^{\text{sparse}}(s_t, s_{t+1}, G) = \begin{cases} 1 & \text{if } D(s_t, G) > D(s_{t+1}, G) \\ -1 & \text{if } D(s_t, G) < D(s_{t+1}, G) \\ 0 & \text{otherwise} \end{cases}$$

As we will show in our empirical ablations, while both $\delta D_{\text{train}}^{\text{sparse}}$ and $\delta D_{\text{train}}^{\text{dense}}$ can be used for conditioning during training, we find the former works better as it does not suffer from scale mismatch at test-time. The pseudocode for reward-conditioning algorithm is given in Appendix A.

## 5 Experiments

We demonstrate the effectiveness of reward conditioning on two recent and representative VLN models: VLN↻BERT and MTVM (Lin et al., 2022b). Additionally, we benchmark our approach against return-conditioning, which has demonstrated strong performance in offline RL tasks.

### 5.1 VLN↻BERT-ORL

VLN↻BERT is a multi-modal transformer model which consists of self-attention and cross-attention layers to fuse the information from visual and text modalities. It implements a recurrent function in V&LBERT (Hao et al., 2020) to capture the entire navigational history in the state token representation which is then used for action prediction. We introduce VLN↻BERT-ORL which refers to VLN↻BERT conditioned on the reward token. For reward conditioning, we add the reward token to the state token at several steps in the pipeline (more details in Appendix B). Since the state token in VLN↻BERT captures navigation history, this allows the model to use the reward token along with navigation history for action prediction. We train the networks from scratch by minimising cross-entropy loss between predicted actions and ground-truth actions in the proposed offline datasets.

### 5.2 Return-conditioned VLN↻BERT

In this approach, instead of conditioning VLN↻BERT on the proposed reward token, we condition it on the returns-to-go token. The conditioning mechanism remains the same, involving addition of the returns-to-go token to the state token at multiple steps in the pipeline. During training, at each timestep, we define the returns-to-go token as the distance of the agent from the goal at that particular step. At test time, since we cannot assume knowledge of the agent's distance from the goal at each step, we initialize the returns-to-go token with the maximum length of trajectories in the validation set. Subsequently, we continuously update it by subtracting the distance traveled by the agent at each step. To ensure a fair comparison, we set the returns-to-go token to zero when the agent reaches close to the goal, under the assumption that the agent can detect its proximity to the goal. We report all the results with the test-time initial return as the maximum length of trajectories in the validation set. However, we also include a performance comparison using the average length of trajectories in the validation set as the initial return in Table 9.

### 5.3 Offline RL Datasets for VLN

Given the lack of any existing studies in offline RL for VLN, we design datasets of varying difficulty levels so as to generate a challenging benchmark for VLN-ORL. We used a pre-trained HAMT agent to generate trajectories for instructions in train sets of R2R and RxR datasets. While it's true that the HAMT model is trained online, its role within our study pertains solely to the generation of an offline dataset. In the context of offline RL, the training process involves utilizing a predetermined set of trajectories, without engaging in online exploration. Thus, our approach aligns with the principles of offline RL, as we exclusively employ the offline dataset for agent training, rather than resorting to online exploration. This distinction remains consistent even in cases where trajectories are sourced from the HAMT model. It is important to emphasize that our core focus lies in advancing offline learning capabilities, irrespective of the trajectory generation method. We will open source our datasets for wider use by the community.

We have created two versions of each dataset D which we generated by rolling out HAMT: 1) D-R2R, generated using train set of R2R, and 2) D-RxR, generated using train set of RxR. Given the instruction and visual state at any step, the model predicts an action which takes the agent from one state to the other. The trajectory ends when either stop action is chosen or maximum time limit is reached. For each instruction in the train set, we generate a trajectory using the pre-trained HAMT model. In addition, we record distance of the agent from goal at every timestep. Specifically, we introduce these five datasets (D):
**Expert:** This dataset consists of the trajectories obtained by rolling out pre-trained HAMT policy.
**15%-noisy data:** It includes trajectories that have been perturbed with 15% noise. That is, at each step, we execute action from HAMT policy with 0.85 probability and a random action otherwise.
**30%-noisy data:** Same as above, except that we raise the noise probability to 30%.
**Mixture:** This dataset consists of a mixture of trajectories from expert and 15% noisy datasets.
**Random:** We also consider a challenging dataset generated by a completely random policy.

### 5.4 Experimental Setup

**Datasets:** We generate several offline datasets using instructions in train sets of R2R and RxR. The R2R dataset has 14,025 instructions in the train set and 4,173 instructions in the test set. The validation set is further divided into val-seen and val-unseen, having 1,020 and 2,349 instructions respectively. We use English subset of RxR which includes 26,464 path-instruction pairs in train set, 2,939 pairs in the val-seen set and 4,551 pairs in the val-unseen set.

**Implementation details:** The experiments were performed on a NVIDIA A100 GPU. We used Adam optimizer with a learning rate of 1e-5 to train the models. The batch size was kept as 64 and the models were trained for 500K iterations. We trained all the models from scratch in the offline RL setup. The image features were extracted using ResNet-152 (He et al., 2016) pre-trained on the Places365 dataset. The model with the best SR on validation unseen set was selected in each case.

**Evaluation Metrics:** We use standard evaluation metrics: (1) Trajectory Length (TL): average length of agent's trajectory (in m) (2) Navigation Error (NE): average distance between the agent's final position and the target (in m); (3) Success Rate (SR): ratio of trajectories reaching the destination with a maximum error of 3 m to the target; (4) Success rate weighted by Path Length (SPL).

### 5.5 Results and Discussion

The performance of VLN↻BERT, return-conditioned and reward-conditioned VLN↻BERT models was evaluated on validation sets of R2R and RxR datasets after being trained on the proposed sub-optimal offline datasets. In particular, the model trained on R2R version of the generated dataset (D-R2R) was tested on the R2R validation set, and likewise for RxR. In the proposed method, we assume that the agent can detect the completion of an instruction. To ensure a fair comparison, the episode of VLN↻BERT is terminated upon reaching the goal state, while the returns-to-go token is set to zero for the return-conditioned agent. Additionally, since the goal states of trajectories in the test set are inaccessible, the reported results do not include performance metrics on the test set. Moving forward, we refer to VLN↻BERT as the baseline model.

Table 1: Performance evaluation of different VLN↻BERT agents on R2R and RxR ORL datasets.

| Data | Methods | R2R | | | | | | | | RxR | | | | | | | |
|---|---|---|---|---|---|---|---|---|---|---|---|---|---|---|---|---|---|
| | | Val seen | | | | Val unseen | | | | Val seen | | | | Val unseen | | | |
| | | TL | NE↓ | SR↑ | SPL↑ | TL | NE↓ | SR↑ | SPL↑ | TL | NE↑ | SR↑ | SPL↑ | TL | NE↓ | SR↑ | SPL↑ |
| + Expert | VLN↻BERT | 10.26 | 5.88 | 59.75 | 58.10 | 9.99 | 6.46 | 54.92 | 53.06 | 16.67 | 10.51 | 44.08 | 42.76 | 15.42 | 9.54 | 47.84 | 45.41 |
| | ReturnC VLN↻BERT | 16.51 | **5.04** | 63.96 | 57.07 | 17.15 | **5.77** | 59.64 | 51.93 | 23.79 | 13.78 | 29.70 | 24.74 | 22.29 | 11.88 | 37.07 | 31.88 |
| | VLN↻BERT-ORL-Dense | 16.85 | 6.60 | 65.92 | 58.9 | 15.65 | 6.46 | 67.01 | 59.74 | 19.84 | 11.05 | 44.27 | 40.06 | 18.93 | 9.59 | 49.44 | 44.84 |
| | VLN↻BERT-ORL-Sparse | 16.17 | 6.96 | **65.92** | **61.21** | 15.28 | 6.94 | **67.09** | **61.36** | 19.94 | **10.16** | **49.10** | **44.50** | 18.57 | 9.11 | **53.11** | **48.16** |
| 15% noisy | VLN↻BERT | 10.24 | 6.17 | 56.22 | 53.99 | 10.58 | 6.80 | 52.11 | 49.51 | 16.13 | 11.13 | 40.39 | 38.46 | 15.23 | 10.19 | 45.18 | 42.86 |
| | ReturnC VLN↻BERT | 14.82 | **5.50** | 58.96 | 54.10 | 15.37 | **6.17** | 57.56 | 51.12 | 23.83 | 13.45 | 31.34 | 26.10 | 22.10 | 12.13 | 36.72 | 31.45 |
| | VLN↻BERT-ORL-Dense | 15.24 | 6.24 | 66.60 | 60.75 | 15.17 | 6.85 | 65.69 | 58.87 | 19.57 | **10.62** | 45.29 | 41.76 | 18.75 | 9.66 | 48.65 | 44.44 |
| | VLN↻BERT-ORL-Sparse | 16.74 | 6.49 | **68.46** | **62.01** | 16.21 | 6.67 | **68.54** | **61.08** | 19.65 | 10.76 | **47.36** | **43.48** | 18.38 | **9.57** | **50.52** | **46.15** |
| 30% noisy | VLN↻BERT | 11.01 | 6.63 | 52.99 | 50.58 | 11.09 | 7.01 | 48.53 | 46.02 | 16.71 | 11.17 | 38.69 | 36.48 | 15.58 | 9.73 | 44.03 | 41.23 |
| | ReturnC VLN↻BERT | 18.21 | **6.56** | 54.65 | 48.56 | 17.74 | **6.66** | 55.34 | 47.2 | 23.22 | 13.25 | 29.87 | 25.97 | 21.99 | 11.42 | 37.02 | 31.58 |
| | VLN↻BERT-ORL-Dense | 17.38 | 7.25 | 61.54 | 54.68 | 15.99 | 6.79 | 65.52 | 57.89 | 20.33 | 11.01 | 41.82 | 38.47 | 19.09 | 9.66 | 49.46 | 45.00 |
| | VLN↻BERT-ORL-Sparse | 16.88 | 7.25 | **63.96** | **57.80** | 15.68 | 6.85 | **67.48** | **60.44** | 20.51 | **10.75** | **46.78** | **42.64** | 19.13 | **9.18** | **51.40** | **46.74** |
| Mixture | VLN↻BERT | 10.43 | 6.21 | 61.21 | 59.27 | 10.27 | 6.46 | 57.17 | 55.02 | 17.01 | 10.68 | 42.50 | 39.87 | 15.76 | 9.38 | 47.44 | 44.66 |
| | ReturnC VLN↻BERT | 15.01 | **5.38** | 61.12 | 56.21 | 15.08 | **5.50** | 61.26 | 55.29 | 21.85 | 10.57 | 34.33 | 29.29 | 20.76 | 9.56 | 38.08 | 32.23 |
| | VLN↻BERT-ORL-Dense | 14.76 | 7.02 | **66.33** | **60.61** | 14.90 | 6.72 | 67.35 | 61.35 | 19.45 | 10.75 | 45.66 | 41.85 | 18.44 | 9.55 | 50.16 | 45.87 |
| | VLN↻BERT-ORL-Sparse | 16.77 | 6.74 | 66.31 | 60.59 | 15.69 | 6.55 | **68.92** | **61.72** | 20.08 | **10.05** | **50.80** | **46.56** | 18.88 | **8.81** | **54.25** | **49.30** |
| Random | VLN↻BERT | 20.86 | 9.94 | 20.18 | 17.02 | 20.78 | 9.48 | 20.48 | 17.87 | 25.47 | 13.59 | 25.86 | 22.12 | 24.05 | 11.80 | 33.09 | 28.69 |
| (100% noisy) | ReturnC VLN↻BERT | 26.98 | 9.76 | 16.36 | 13.64 | 27.30 | 9.37 | 14.13 | 11.07 | 25.86 | 13.60 | 23.48 | 19.13 | 24.77 | 12.41 | 25.99 | 21.70 |
| | VLN↻BERT-ORL-Dense | 18.83 | **7.63** | 56.81 | 50.74 | 18.94 | 7.87 | 56.96 | 48.65 | 23.01 | **12.09** | 36.03 | 31.84 | 21.97 | 10.95 | 39.24 | 34.65 |
| | VLN↻BERT-ORL-Sparse | 19.24 | 7.71 | **57.88** | **50.79** | 18.44 | **7.46** | **60.66** | **51.11** | 22.54 | 12.25 | **38.52** | **34.89** | 20.87 | **10.79** | **43.35** | **38.62** |

The results, which are showcased in Table 1, indicate that reward-conditioned VLN↻BERT outperforms both the baseline and the return-conditioned model on all offline datasets, including Expert, Noisy, Random, and Mixture. The reward-conditioned model achieved a success rate that was at least 5% higher than the baseline on both R2R and RxR validation sets, even when trained on the Expert dataset. Moreover, the reward-conditioned model demonstrated a significant improvement on R2R validation unseen set, achieving around 13% higher success rate compared to the baseline. Furthermore, the reward-conditioned model achieved around 20% higher success rate than the return-conditioned model on the validation seen set of RxR. The improvement due to reward-conditioning was even more pronounced when the models were trained on Noisy datasets. Specifically, the reward-conditioned model achieved a success rate about 15% higher than the baseline on the R2R validation sets and around 5% higher on the RxR validation sets when trained on the 15% Noisy dataset. Compared to the return-conditioned model, the improvement was around 10% on the R2R validation sets and around 15% on the RxR validation sets.

Notably, when the models were trained on 30% Noisy datasets, the reward-conditioned agent achieved a 20% higher success rate as compared to the baseline on the validation unseen set of R2R. On the RxR validation sets, the reward-conditioned agent achieved around 15% higher success rates than the return-conditioned agent. Furthermore, when trained on the Random dataset, the reward-conditioned model achieved a roughly 40% higher success rate on the R2R validation sets compared to both the baseline and the return-conditioned agents. This demonstrates that the reward-conditioned model is robust to sub-optimal training datasets.

Moreover, the results show that addition of noise negatively impacted the performance of both VLN↻BERT and return-conditioned models, while the performance of reward-conditioned model was not significantly affected. This is also evident from Fig. 2(c)(d), which clearly illustrates that as noise is introduced, the performance gap between the reward-conditioned model and the baseline widens. The model performance on the Mixture dataset was slightly better than that on the Expert dataset, possibly due to inclusion of 15%-noisy trajectories along with expert trajectories, providing a more diverse set of experiences for training. Furthermore, we note that the reward-conditioned model occasionally demonstrates a higher navigation error than the baseline. This discrepancy can be attributed to our model being conditioned on a zero reward only when it reaches proximity to the goal. In cases where the agent fails to reach the goal, the conditioning on a +1 reward incentivizes continuous movement, leading to both higher navigation errors and increased trajectory lengths in certain instances. Overall, our evaluation clearly shows that offline RL via reward conditioning is an effective approach to improve the performance of VLN agents on suboptimal datasets. It enables the model to learn from noisy datasets and achieve superior results, especially in challenging scenarios such as on the Random dataset.

Interestingly, while the return-conditioned model outperforms the baseline on most of the R2R datasets, it demonstrates less effectiveness on the RxR datasets. This discrepancy may be attributed to the significant variance in trajectory lengths within the RxR dataset, which hinders generalization when initializing the test-time returns based on the maximum trajectory length. Consequently, this results in a decline in performance, highlighting the limitations of return-conditioning in scenarios where test-time returns are un-

known. Furthermore, as previously discussed, we present the performance comparison of return-conditioned VLN↻BERT on RxR datasets with different test-time initial returns in Table 9. The results illustrate that using the maximum length of trajectories as the initial return yields superior performance compared to using the average length.

**Why is the reward-conditioned model able to learn from suboptimal datasets?**

We have a reward $r_t$ associated with each action $a_t$ in the dataset, defined as:

$$r_t = \delta D_{\text{t, train}}^{\text{dense}}(s_t, s_{t+1}, G) = D(s_t, G) - D(s_{t+1}, G)$$

Here, $t$ represents the time step, and $t \geq 0$. The resulting value of $\delta D_{\text{t, train}}^{\text{dense}}(s_t, s_{t+1}, G)$ provides valuable insights: a positive value signifies that the agent is making progress towards the goal, while a negative value indicates movement away from the goal. A value of zero represents no change, which occurs when the agent has successfully reached the goal.

Now, in the context of the suboptimal dataset, certain actions contribute to the agent's progression towards the goal (yielding positive rewards), while others cause the agent to move away from the goal (leading to negative rewards). At a given time step t, let's say a specific trajectory involves action $a_t$ associated with reward $r_t$. The model $M$ which predicts actions would be conditioned on this reward $r_t$ at that timestep. Consequently, the model $M$ knows that it must generate an action that would lead to an change in distance indicated by $r_t$. If the action $a_t$ is suboptimal—potentially causing the agent to move away from the goal, even when the optimal choice is to move towards it—the corresponding reward $r_t$ would be negative. Conditioning model $M$ on this negative reward effectively instructs it to generate an action that aligns with the suboptimal behavior demonstrated in the dataset. Paradoxically, this doesn't hinder the model's learning as it already knows that it needs to output an suboptimal action (indicated by a negative reward) at that timestep. Thus, minimizing the disparity between predicted and actual actions makes the model grasp the connection between actions and their associated rewards.

In essence, through this reward-guided training process, the model acquires the ability to predict actions contingent upon reward information. This dynamic becomes particularly advantageous during testing. At test time, we can condition the model with positive rewards at each timestep. As a result, the model is primed to consistently generate actions that propel the agent closer to the goal. This predictive behavior is a direct result of the model's training, as it has learned to associate positive rewards with actions that effectively drive progress towards the goal.

## 5.6 Ablation studies

**Is the proposed reward token too greedy?:** It might seem that the reward token used for conditioning the model is too greedy and can lead the agents to get stuck in cases where they need to consider the long-term impact of their actions. To address this concern, we conducted a comprehensive analysis of the evaluation sets to gain deeper insights into the effectiveness of our approach. More specifically, we examined the trajectories within our evaluation sets to determine the percentage of cases where the agent must deviate from the intended goal at least once to eventually reach it, indicating a capacity to reason about the long-term consequences of its actions. We observed that within the R2R validation sets, this scenario is relatively infrequent. However, within the RxR validation seen and unseen sets, a substantial portion—54.2% and 55.11% respectively—of trajectories fall under this category. We undertook a separate evaluation of our approach specifically on these types of trajectories. To achieve this, we partitioned each of the RxR validation sets (both the "val seen" and "val unseen" sets) into two distinct subsets. The first subset, referred to as "val tough", comprises trajectories that involve deviations from the goal at least once. On the other hand, the second subset, termed as "val easy", includes trajectories that do not involve any such deviations. In order to assess the effectiveness of our approach, we measured success rates (SR) and success rates weighted by path length (SPL) on all four of these subsets: "val tough" and "val easy" within both the "val seen" and "val unseen" sets of RxR. The results are shown in Table 2.

Table 2: Performance evaluation of VLN↻BERT agents on different subsets of RxR validation sets.

| Datasets | Methods | Val seen tough | | Val seen easy | | Val unseen tough | | Val unseen easy | |
|---|---|---|---|---|---|---|---|---|---|
| | | SR ↑ | SPL ↑ | SR↑ | SPL↑ | SR↑ | SPL ↑ | SR↑ | SPL↑ |
| Expert | VLN↻BERT | 36.72 | 33.15 | 53.34 | 52.01 | 36.12 | 33.24 | 62.31 | 60.28 |
| | ReturnC VLN↻BERT | 21.91 | 17.31 | 39.00 | 33.59 | 29.35 | 24.38 | 46.74 | 41.14 |
| | VLN↻BERT-ORL-Sparse | **41.12** | **36.42** | **61.07** | **57.47** | **41.63** | **36.33** | **68.38** | **64.45** |
| 15% noisy | VLN↻BERT | 31.58 | 28.9 | 52.23 | 50.63 | 33.53 | 30.55 | 58.39 | 56.77 |
| | ReturnC VLN↻BERT | 22.91 | 18.66 | 41.53 | 35.07 | 26.59 | 22.09 | 49.44 | 43.16 |
| | VLN↻BERT-ORL-Sparse | **38.92** | **34.76** | **56.98** | **52.83** | **39.95** | **35.14** | **66.62** | **61.89** |
| 30% noisy | VLN↻BERT | 26.49 | 24.67 | 44.21 | 42.73 | 28.51 | 25.93 | 52.72 | 51.03 |
| | ReturnC VLN↻BERT | 21.85 | 18.02 | 39.38 | 35.38 | 28.35 | 23.38 | 47.72 | 41.68 |
| | VLN↻BERT-ORL-Sparse | **33.71** | **30.14** | **51.78** | **47.88** | **37.4** | **32.23** | **62.8** | **58.56** |
| Random | VLN↻BERT | 14.79 | 13.92 | 29.84 | 27.08 | 23.17 | 20.24 | 38.4 | 36.59 |
| | ReturnC VLN↻BERT | 17.45 | 13.46 | 30.16 | 25.28 | 18.06 | 14.79 | 36.81 | 31.01 |
| | VLN↻BERT-ORL-Sparse | **28.69** | **25.04** | **46.06** | **41.96** | **31.66** | **26.75** | **55.41** | **50.14** |

Table 3: K-fold validation results for VLN↻BERT agents trained on different subsets of the 30% Noisy R2R dataset.

| %dataset | Methods | R2R val seen | | R2R val unseen | |
|---|---|---|---|---|---|
| | | Average Accuracy | $\sigma$ | Average Accuracy | $\sigma$ |
| 25% | VLN↻BERT | 38.23 | 3 | 37.55 | 2.88 |
| | VLN↻BERT-ORL-Sparse | 54.64 | 1.71 | 54.87 | 2.32 |
| 50% | VLN↻BERT | 44.10 | 2.61 | 42.02 | 2.35 |
| | VLN↻BERT-ORL-Sparse | 59.97 | 1.36 | 60.17 | 2.05 |
| 75% | VLN↻BERT | 48.88 | 1.74 | 45.28 | 2.1 |
| | VLN↻BERT-ORL-Sparse | 63.01 | 0.91 | 64.34 | 1.35 |

It is evident from the results that reward conditioning improves performance across all four subsets, notably even in more challenging "val tough" scenarios. When trained on the Expert dataset, the reward conditioning approach yields a 5% higher success rate for both seen and unseen "val tough" subsets compared to the baseline. Additionally, it demonstrates a significant improvement of around 20% compared to the return-conditioned agent on most of the subsets. The improvement was even more pronounced when the models were trained on the Noisy datasets. When trained on the 15% Noisy dataset, reward conditioning led to roughly a 7% enhancement compared to the baseline and approximately a 15% improvement compared to the return-conditioned agent on the "val tough" subsets of both seen and unseen validation sets. Furthermore, when the model was trained on the 30% Noisy dataset, the reward-conditioned agent achieved a 9% higher success rate on the "val unseen tough" set compared to the baseline. Even when trained on the Random dataset, the reward-conditioned model showed exceptional performance gains, achieving a 14% success rate improvement for the "val seen tough" set and an 8% improvement for the "val unseen tough" set compared to the baseline. Additionally, a noteworthy boost of approximately 17% in performance is observed for the "val easy" subsets of both "val seen" and "val unseen" sets compared to the baseline. Moreover, compared to the return-conditioned agent, the reward-conditioned agent demonstrated a 15% higher success rate on the "val seen easy" set. Although the proposed reward token seems to be greedy, it effectively enhances performance across diverse scenarios, including those requiring deviations from the intended goal.

**Is it really not too greedy?:** While "val-tough" accounts for trajectories with at least one deviation from the goal, it remains uncertain whether the proposed reward-conditioned agent will excel with a higher number of consecutive deviations from the goal. To assess the robustness of our model, we extended our analysis to explore trajectories where the agent consecutively deviates from the goal at least twice, denoted as N2. Additionally, we examined cases of consecutive deviations denoted as N3, N4, and N5, where Ni represents the number of trajectories where the agent moves away from the goal consecutively at least i times. The analysis, shown in Table 8, reveals a substantial number of trajectories involving consecutive deviations of the agent from the goal. In this table, N0 represents number of trajectories which do not involve any deviation from the goal. Let Ti denote the set of trajectories where the agent moves away from the goal atleast i times consecutively. We assessed the performance of the proposed method, the baseline agent and the return-conditioned model on each of these sets (Ti) across both validation seen and unseen sets of RxR. We report the SR and SPL for each set in the Tables 6 and 7.

Table 4: Impact of training seed variation on performance of VLN↻BERT agents on 30% Noisy R2R dataset.

| %dataset | Methods | R2R val seen | | R2R val unseen | |
|---|---|---|---|---|---|
| | | Average Accuracy | $\sigma$ | Average Accuracy | $\sigma$ |
| 25% | VLN↻BERT | 39.13 | 1.03 | 37.93 | 0.17 |
| | VLN↻BERT-ORL-Sparse | 53.94 | 0.34 | 55.39 | 0.51 |
| 50% | VLN↻BERT | 43.8 | 0.64 | 41.47 | 0.11 |
| | VLN↻BERT-ORL-Sparse | 59.05 | 1.01 | 59.18 | 0.51 |
| 75% | VLN↻BERT | 47.41 | 0.2 | 44.81 | 0.15 |
| | VLN↻BERT-ORL-Sparse | 62.53 | 0.3 | 63.87 | 0.26 |

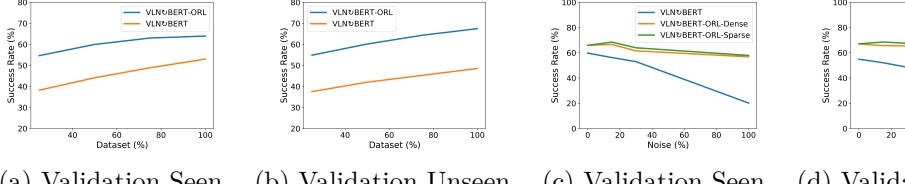

(a) Validation Seen    (b) Validation Unseen    (c) Validation Seen    (d) Validation Unseen

Figure 2: Performance comparison of VLN↻BERT and reward-conditioned VLN↻BERT on the validation sets (a, b) based on varying sizes of the training subset from the 30% Noisy R2R dataset (c, d) as a function of the level of noise in the proposed R2R training dataset.

As evident from the data presented in the tables, employing reward-conditioning yields performance enhancements, even in scenarios involving consecutive deviations from the goal. When the models are trained on the Expert dataset, the utilization of reward-conditioning results in a performance boost of around 5% on the T2 set within the validation unseen set as compared to the baseline. Moreover, this enhancement extends to other sets (Ti) within the validation seen and unseen sets, with performance improvements ranging from 3% to 5%. Compared to the return-conditioned model, the reward-conditioned model exhibits approximately a 15% higher success rate on most of the Ti sets within validation seen. When trained on 15% Noisy dataset, reward-conditioning enhances performance by around 4-5% across most sets compared to the baseline.

Upon training on the 30% Noisy dataset, the efficacy of reward-conditioning becomes more pronounced, yielding an approximate 7% improvement in performance on the T2 sets in both validation seen and unseen sets. Across other sets (Ti), performance enhancements of approximately 3% to 7% are observed. Moreover, compared to the return-conditioned agent, the performance improves by around 10% on most subsets of the RxR validation seen set. When trained on the Random dataset, the reward-conditioned agent achieves a 14% higher success rate compared to baseline on the T4 set of validation unseen. Substantial improvements, around 10%, are observed on most other sets as well. Additionally, the performance improves by around 12% on most subsets of validation unseen set compared to the return-conditioned agent. Despite the inherent greediness of the proposed reward token, it remarkably boosts performance across various scenarios, including those necessitating consecutive deviations from the intended goal.

However, it's crucial to acknowledge that our reward-conditioned model exhibits a decline in performance when faced with sets characterized by a higher number of consecutive deviations. This could be attributed to several factors. Primarily, these sets inherently comprises more intricate and challenging scenarios, thereby presenting a more formidable testing ground. Additionally, the model's inherent greediness might occasionally pose limitations, especially in scenarios where optimal navigation involves substantial deviations from the goal before convergence. However, the empirical evidence we have presented in our analysis conclusively demonstrates the efficacy of our approach in improving performance even within complex scenarios, without entailing the risk of agents getting stuck. Thus, we feel it is reasonable to anticipate that the efficacy of the proposed greedy reward conditioning technique would persist even in intricate and complex environments.

**Size of the dataset:** We conducted an evaluation of the performance of VLN↻BERT and reward-conditioned VLN↻BERT trained on different subsets of the 30% Noisy R2R dataset. The results of our evaluation, as shown in Fig. 2(a)(b), indicate that the performance of both models improves as the dataset

Table 6: Performance evaluation of VLN↻BERT agents on different subsets of RxR validation seen set.

| Datasets | Methods | T2 | | T3 | | T4 | | T5 | |
|---|---|---|---|---|---|---|---|---|---|
| | | SR | SPL | SR | SPL | SR | SPL | SR | SPL |
| Expert | VLN↻BERT | 31.62 | 28.06 | 25.43 | 25.81 | 27.11 | 24.77 | 24.52 | 22.39 |
| | ReturnC VLN↻BERT | 16.19 | 12.4 | 16.9 | 13.78 | 14.21 | 10.82 | 15.14 | 14.33 |
| | VLN↻BERT-ORL-Sparse | **34.31** | **30.44** | **28.48** | **28.69** | **29.11** | **27.16** | **27.71** | **24.42** |
| 15% noisy | VLN↻BERT | 26.21 | 22.87 | 21.76 | 19.07 | 17.13 | 15.41 | 17.05 | 15.91 |
| | ReturnC VLN↻BERT | 16.43 | 13.56 | 17.86 | 14.84 | 14.08 | 11.46 | 11.43 | 10.17 |
| | VLN↻BERT-ORL-Sparse | **30.36** | **26.35** | **26** | **22.97** | **21.72** | **17.93** | **21.98** | **18.22** |
| 30% noisy | VLN↻BERT | 23.93 | 21.68 | 20.95 | 19.12 | 15.9 | 15.24 | 15.14 | 14.36 |
| | ReturnC VLN↻BERT | 15.71 | 13.22 | 13.48 | 12.37 | 9.63 | 8.4 | 8.71 | 8.71 |
| | VLN↻BERT-ORL-Sparse | **30.71** | **26.34** | **26.43** | **22.92** | **19.72** | **18.04** | **18.1** | **17.58** |
| Random | VLN↻BERT | 10.14 | 9.67 | 8.57 | 8.12 | 8.45 | 8.04 | 6.43 | 6.6 |
| | ReturnC VLN↻BERT | 12.38 | 9.24 | 11.9 | 9.24 | 8.45 | 6.4 | 10.19 | 9.75 |
| | VLN↻BERT-ORL-Sparse | **22.98** | **19.72** | **19.05** | **16.21** | **15.02** | **12.8** | **14.29** | **11.94** |

Table 7: Performance evaluation of VLN↻BERT agents on different subsets of RxR validation unseen set.

| Datasets | Methods | T2 | | T3 | | T4 | | T5 | |
|---|---|---|---|---|---|---|---|---|---|
| | | SR | SPL | SR | SPL | SR | SPL | SR | SPL |
| Expert | VLN↻BERT | 30.55 | 27.15 | 26.47 | 23.8 | 23.6 | 21.2 | 17.39 | 14.36 |
| | ReturnC VLN↻BERT | 22.86 | 19.71 | 20.85 | 18.7 | 15.34 | 13.94 | 13.04 | 12.19 |
| | VLN↻BERT-ORL-Sparse | **35.16** | **30.89** | **29.36** | **26.53** | **26.55** | **24.74** | **21.01** | **19.27** |
| 15% noisy | VLN↻BERT | 28.48 | 26.16 | 25.75 | 23.87 | 19.35 | 17.78 | 14.94 | 14.07 |
| | ReturnC VLN↻BERT | 20.81 | 17.75 | 17.42 | 15.67 | 12.98 | 11.04 | 10.87 | 10.23 |
| | VLN↻BERT-ORL-Sparse | **33.48** | **29.97** | **29.9** | **26.53** | **24.78** | **22.24** | **18.12** | **15.55** |
| 30% noisy | VLN↻BERT | 24.11 | 20.17 | 20.77 | 17.83 | 15.76 | 14.62 | 9.32 | 8.07 |
| | ReturnC VLN↻BERT | 20.76 | 19.77 | 18.42 | 17.88 | 11.53 | 10.56 | 10.22 | 10.78 |
| | VLN↻BERT-ORL-Sparse | **31.43** | **26.83** | **25.79** | **22.42** | **20.47** | **18.46** | **13.77** | **12.87** |
| Random | VLN↻BERT | 14.23 | 12.36 | 9.82 | 9.15 | 6.31 | 6.01 | 4.35 | 4.35 |
| | ReturnC VLN↻BERT | 12.23 | 10.41 | 9.6 | 8.37 | 7.67 | 6.59 | 6.52 | 6.52 |
| | VLN↻BERT-ORL-Sparse | **24.62** | **20.95** | **22.77** | **19.2** | **20.06** | **16.33** | **13.04** | **10.16** |

size increases. Initially, the improvement is substantial, but it decreases gradually as we increase the subset size of the dataset. This can be attributed to the diminishing marginal benefit of adding new trajectories to the dataset. Since the model may have already encountered similar trajectories before, the value of adding new ones decreases over time.

We further run k-fold validation by randomly selecting x% of the dataset, k=3 times. In Table 3, we report the average accuracy and standard deviation ($\sigma$). Notably, the reward-conditioned model demonstrates lower prediction variance, indicating that reward-conditioning contributes to the development of a more robust model, one that is less susceptible to the specific data partitions.

**Training seed variation:** We assessed the performance of VLN↻BERT and reward-conditioned VLN↻BERT across different subsets of the 30% Noisy R2R dataset using three distinct model seeds. In each instance, we maintain the data subset fixed and train the model using three different seeds on that particular dataset. The results are shown in Table 4. Notably, the low standard deviations suggest that the model's performance exhibits minimal variation with changes in the random seed, highlighting its robustness and stability across different initial conditions.

**Sparse vs Dense Rewards:** To gain further insights into our model's performance, we conducted a more detailed analysis by considering dense and sparse reward tokens during training. At test time, we condition the model on sparse reward tokens only. Table 1 shows that the sparse-reward conditioned model performs slightly better than the dense-reward conditioned model. One possible explanation for this is that the sparse reward case emulates the test conditions during training, which aids the model in performing better at test time. This experiment highlights that reward-conditioning substantially enhances performance, even when distances from the goal location are not known at every step. It is only necessary to have knowledge of goal when the agent reaches close to it. We propose that this information can be readily provided to the agent by placing a beeper at the goal.

**How do we condition?:** We explore two conditioning approaches for VLN↻BERT: concatenation and addition of reward token to state representation. Section 5.1 discusses addition technique. In concatenation approach, we project reward token into an embedding space of size one-fourth the dimension of common embedding space. This projected embedding is then concatenated with state representation, followed by a

linear layer to restore original embedding size. This form of reward concatenation is applied at every step where reward addition is performed in Section 5.1. We train both models on 30% Noisy R2R dataset and observe that they perform similarly on validation sets, as depicted in Table 5. These results indicate that both conditioning techniques can provide necessary conditional information.

**Alternative baseline model:** We further extend our analysis to include the conditioning of a more recent VLN model, MTVM (Lin et al., 2022b). The architectural details are discussed in Appendix B. The results presented in Table 10 clearly demonstrate the effectiveness of reward-conditioning in enhancing performance across all proposed datasets. Even on the Expert dataset, reward-conditioned model showcases a remarkable improvement of approximately 20%. Furthermore, the performance gap between the reward-conditioned model and the baseline consistently widens as dataset's noise level increases. Notably, on the challenging Random dataset, the reward-conditioned model exhibits a success rate that

Table 5: Performance of different reward conditioning schemes; on the 30% Noisy R2R dataset.

| Methods | R2R | | | |
|---|---|---|---|---|
| | Val seen | | | |
| | TL | NE ↓ | SR ↑ | SPL ↑ |
| Concatenation | 16.81 | **6.81** | **64.64** | **59.68** |
| Addition | 16.88 | 7.25 | 63.96 | 57.80 |
| Methods | R2R | | | |
| | Val unseen | | | |
| | TL | NE ↓ | SR ↑ | SPL ↑ |
| Concatenation | 16.66 | **6.76** | 66.67 | 58.43 |
| Addition | 15.68 | 6.85 | **67.48** | **60.44** |

is approximately 30% higher than the baseline. These findings underscore the significant performance enhancement achieved by reward-conditioning in MTVM on suboptimal datasets. Since the proposed offline RL method also yielded positive results for VLN↺BERT, its generalizability holds great promise for improving performance of VLN agents.

Table 8: Analysis of the trajectories in the validation sets of RxR.

| Set | Total | N0 | N1 | N2 | N3 | N4 | N5 |
|---|---|---|---|---|---|---|---|
| Val seen | 2939 | 1346 | 1593 | 840 | 420 | 213 | 105 |
| Val unseen | 4551 | 2043 | 2508 | 1365 | 729 | 339 | 138 |

Table 9: Performance comparison of return-conditioned VLN↺BERT on RxR datasets with different test-time initial returns.

| Data | Initial return | Val seen | | | | Val unseen | | | |
|---|---|---|---|---|---|---|---|---|---|
| | | TL | NE | SR | SPL | TL | NE | SR | SPL |
| Expert | Average | 14.47 | **9.55** | **30.83** | **26.93** | 14.06 | **8.92** | 35.68 | 31.65 |
| | Maximum | 23.79 | 13.78 | 29.7 | 24.74 | 22.29 | 11.88 | **37.07** | **31.88** |
| 30% Noisy | Average | 16.11 | **9.91** | 28.62 | **26.04** | 15.85 | **9.12** | 35.79 | 30.93 |
| | Maximum | 23.22 | 13.25 | **29.87** | 25.97 | 21.99 | 11.42 | **37.02** | **31.58** |
| Random | Average | 23.92 | 13.8 | 9.49 | 6.85 | 23.49 | **12.36** | 15.58 | 12.25 |
| | Maximum | 25.86 | **13.6** | **23.48** | **19.13** | 24.77 | 12.41 | **25.99** | **21.7** |

Table 10: Performance evaluation of different MTVM agents on ORL datasets for R2R.

| Data | Methods | R2R | | | | | | | |
|---|---|---|---|---|---|---|---|---|---|
| | | Val seen | | | | Val unseen | | | |
| | | TL | NE ↓ | SR ↑ | SPL ↑ | TL | NE ↓ | SR ↑ | SPL ↑ |
| Expert | MTVM | 12.77 | **6.59** | 46.82 | 38.52 | 12.92 | **6.01** | 43.59 | 38.52 |
| | MTVM-ORL-Dense | 17.51 | 6.63 | 64.84 | 57.25 | 16.98 | 6.50 | 65.90 | **60.57** |
| | MTVM-ORL-Sparse | 16.39 | 6.73 | **65.23** | **60.91** | 15.91 | 6.75 | **66.84** | 59.66 |
| 15% noise | MTVM | 11.12 | **6.41** | 40.74 | 37.95 | 11.25 | **6.45** | 41.63 | 38.22 |
| | MTVM-ORL-Dense | 17.51 | 7.05 | 61.12 | 56.67 | 16.79 | 6.87 | 62.98 | 56.67 |
| | MTVM-ORL-Sparse | 16.66 | 6.65 | **66.21** | **60.43** | 16.22 | 6.77 | **66.84** | **58.66** |
| 30% noise | MTVM | 9.00 | 7.70 | 36.74 | 35.9 | 9.04 | 28.5 | 36.87 | 39.8 |
| | MTVM-ORL-Dense | 17.69 | 7.61 | 58.18 | 52.89 | 16.63 | 7.23 | 61.6 | 53.97 |
| | MTVM-ORL-Sparse | 17.51 | **6.54** | **65.13** | **59.15** | 16.79 | **6.24** | **67.18** | **58.25** |
| Mixture | MTVM | 9.35 | **6.04** | 52.69 | 51.78 | 9.23 | **6.29** | 52.19 | 50.34 |
| | MTVM-ORL-Dense | 16.81 | 6.61 | 65.52 | 59.61 | 16.52 | 6.88 | 65.56 | 58.48 |
| | MTVM-ORL-Sparse | 16.60 | 6.28 | **68.46** | **62.92** | 16.39 | 6.79 | **67.18** | **59.94** |
| Random | MTVM | 24.43 | 9.51 | 25.56 | 19.99 | 24.36 | 8.65 | 33.33 | 23.33 |
| | MTVM-ORL-Dense | 20.58 | 7.77 | 53.87 | 45.33 | 20.57 | **7.50** | 54.96 | 44.35 |
| | MTVM-ORL-Sparse | 20.11 | **8.93** | **54.55** | **47.75** | 19.41 | 7.74 | **58.32** | **48.19** |

## 6 Conclusion and Limitations

We introduced the setting of offline RL for VLN. We showed that conditioning the model on a suitably shaped reward token can significantly improve performance of VLN agents trained on large suboptimal

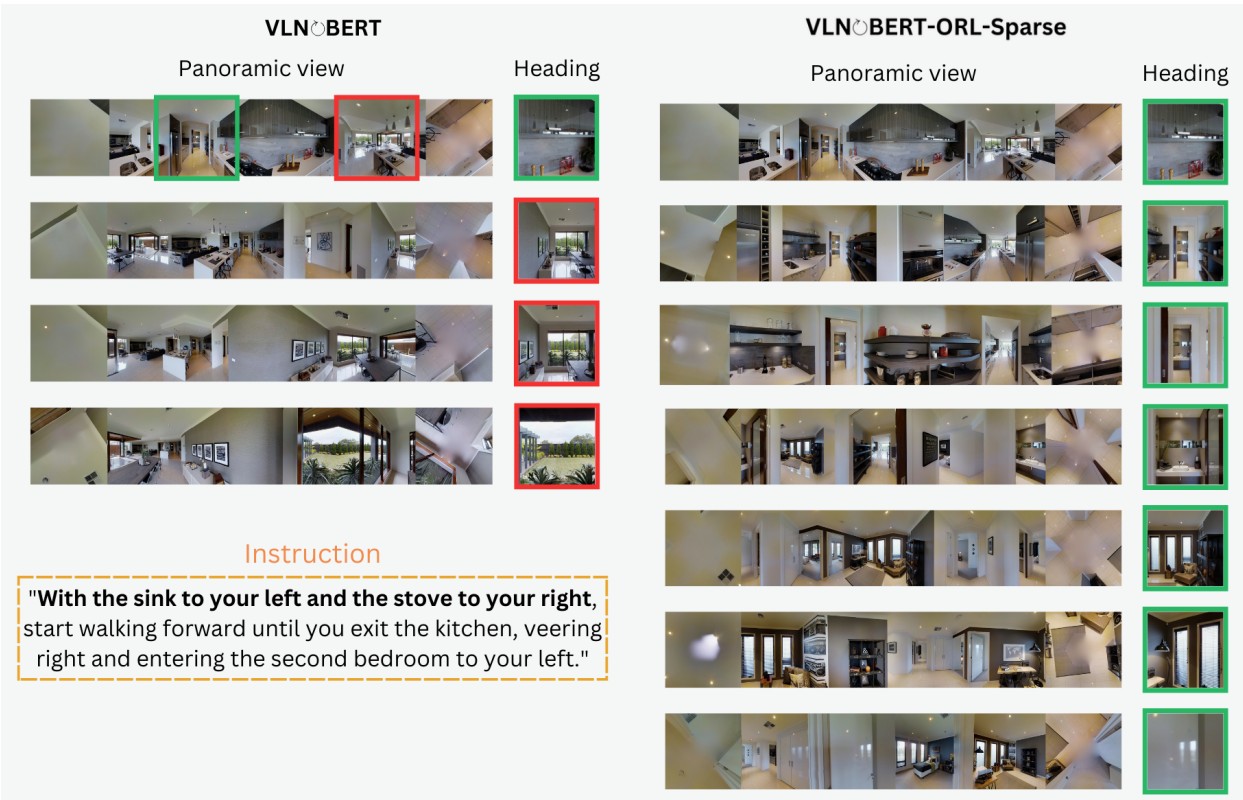

Figure 3: Visualisation of panoramic views and headings (view in heading direction) at every step for agents trained on the 30% Noisy R2R dataset. The reward-conditioned agent correctly follows the instruction to reach the goal whereas the baseline agent takes the wrong path and reaches elsewhere. Instead of moving forward with the sink to the left and stove to right (indicated by green box) the baseline agent goes in the other direction to the hall (indicated by red box). It continues in that direction and eventually exits the house. Conversely, the reward-conditioned agent correctly navigates the route, exiting the kitchen and entering the bedroom as instructed.

datasets. Further, we introduced several benchmarks for VLN-ORL to promote research in this area and conducted extensive ablations on the design of reward tokens. While our method has exhibited superior performance on various suboptimal datasets, it is important to acknowledge some limitations. Firstly, our approach relies on prior knowledge of stop state to terminate episodes, and we plan to address this limitation in our future work. Secondly, encouraging the agent to move closer to the goal at each step may result in longer paths in certain scenarios, and we will investigate strategies to mitigate this effect. Furthermore, we aim to explore alternative offline RL algorithms and architectures, including value-based methods and diffusion-based policies, to broaden the scope of our research and enhance performance in VLN tasks.

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

## A    Methodology

Algorithm 1 describes reward-token conditioning in detail.

---
**Algorithm 1: Reward token conditioning**

---
*Training Phase*:

**Input:** Instruction $I$, Visual features $V_t$, State token $q_{t-1}$, Ground-truth action $a_t$, Current state $s_t$,
  Next state $s_{t+1}$, Goal location G

**Output:** Trained policy model M parameters

**1** At any timestep t, predict action by conditioning M on the reward token

$a_t^{'} = M(q_{t-1}, I, V_t, \delta D_{\text{train}}^{\text{sparse}}(s_t, s_{t+1}, G))$

**2** Take ground-truth action $a_t$ to go to next state $s_{t+1}$

**3** Compute cross-entropy loss between the predicted action $a_t^{'}$ and the ground-truth action $a_t$

**4** Optimize loss until convergence to train M

*Testing Phase*:

**Input:** Instruction $I$, Visual state $V_t$, Current state $s_t$, State token $q_{t-1}$

**Output:** Next state $s_{t+1}$

**5** At any timestep t, use the trained model M to predict action

$a_t^{'} = M(I, V_t, q_{t-1}, \delta D_{\text{test}}^{\text{sparse}})$

**6** Execute the action $a_t^{'}$ to move to the new state $s_{t+1}$

---

## B    Experiments

### B.1    Model Details

**1. VLN↻BERT:** We use the architecture of VLN↻BERT Hong et al. (2021) as our baseline agent policy. VLN↻BERT is a multi-modal transformer model which consists of self-attention and cross-attention layers to fuse the information from visual and text modalities. It implements a recurrent function in V&LBERT Hao et al. (2020) to capture the entire navigational history in the state representation. At each timestep, the agent receives three inputs, the previous state token $q_{t-1}$, the language instruction $I$ and the visual features $V_t$ for the current scene. VLN↻BERT employs a self-attention mechanism that operates on the cross-modal tokens, allowing it to capture the correlation between the textual and visual inputs. This enables the model to determine the action probabilities $p_t^a$ accurately at each timestep $t$.

$$q_t, p_t^a = \text{VLN} \circlearrowleft \text{BERT}(q_{t-1}, I, V_t)$$

Initially, the language instruction is fed into VLN↻BERT and the linguistic features are extracted. The final hidden state corresponding to the predefined [CLS] token is taken as the aggregate sequence representation. It is also used as the initial state representation $q_0$. At every timestep, the agent observes a panoramic view of the environment consisting of 36 images, $\{v_t^i\}_{i=1}^{36}$. The visual features $V_t$ are extracted using ResNet-152 He et al. (2016) pre-trained on the Places365 dataset Zhou et al. (2017). A vision encoder $F_v$ is then used to project the extracted features into the common embedding space.

$$V_t^{'} = F_v(V_t)$$

The visual tokens $V_t^{'}$ are concatenated with the state token $q_{t-1}$ to obtain the state-visual features $H_t$. The state-visual representation is then updated by calculating cross-modal attention between the state-visual features $H_t$ and the instruction features $f_i$.

$$H_t^{'} = \text{CrossAttention}(H_t, f_i, \theta_c)$$

where $\theta_c$ are the parameters of the cross-attention module. The updated state-visual features $H_t^{'}$ are then fed to a self-attention module to capture the relationship between the state and the visual features.

$$H_t^{''}, p_t^a = \text{SelfAttention}(H_t^{'}, \theta_s)$$

where $\theta_s$ represents the self-attention module parameters. The state-visual features and the instruction features are passed through the cross-attention and self-attention modules multiple times to refine the representations and capture state-visual relationships. The similarity scores $p_t^a$ computed between the state representation and the visual features in the last self-attention module are used as the action prediction probabilities. The action predicted with the highest probability is chosen at each timestep.

**2. VLN↺BERT-ORL:** To condition VLN↺BERT on rewards, we add the reward token to the state token at several steps. Since the state token captures navigation history, this allows the model to use the reward token along with the navigation history to correctly predict the action. We firstly extract the instruction features $f_i$, the visual features $V_t^{'}$ and the state representation $q_{t-1}$. Thereafter, we add the reward token $\delta D_{\text{train, t}}^{\text{sparse}}$ to the state representation and concatenate the updated state representation with the visual tokens to obtain the state-visual features $H_t^{'}$. Next, cross-attention is performed between these features, with the state-visual features serving as queries and the instruction features serving as both keys and values.

$$q_{t-1}^{'} = q_{t-1} + \delta D_{\text{train, t}}^{\text{sparse}}$$

$$H_t^{'} = \text{Concat}(q_{t-1}^{'}, V_t^{'})$$

$$H_t^{''} = \text{CrossAttention}(H_t^{'}, f_i, \theta_c)$$

$$q_{t-1}^{''} = H_t^{''}[0]$$

$$V_t^{''} = H_t^{''}[1:]$$

In order to strengthen the incorporation of conditioning information by the model, we further add the reward token $\delta D_{\text{train, t}}^{\text{sparse}}$ to the state token $q_{t-1}^{''}$. The updated state-visual representation is then fed to the self-attention module. This approach ensures that the reward token is given significant consideration by the model during the decision-making process.

$$q_{t-1}^{'''} = q_{t-1}^{''} + \delta D_{\text{train, t}}^{\text{sparse}}$$

$$H_t^{'''} = \text{Concat}(q_{t-1}^{'''}, V_t^{''})$$

$$H_t^{''''}, p_t^a = \text{SelfAttention}(H_t^{'''}, \theta_a)$$

The VLN↺BERT architecture is composed of several blocks, each of which includes the cross-attention and self-attention modules. We incorporate the conditioning information into the state token in each block. In the final block, action prediction probabilities $p_t^a$ are obtained by normalizing the attention scores between the state token and the visual features. Fig. 4 depicts the reward token conditioned model pipeline.

Formally, we minimise the navigation loss, which is,

$$L = -\sum_t a_t^* \log p_t^a$$

where $a_t^*$ refers to the teacher action and $p_t^a$ denotes the action prediction probabilities at timestep t.

**3. MTVM-ORL:** MTVM consists of a language encoder, a vision encoder and a cross-modality encoder. Additionally, it also stores activations of the previous time steps in a memory bank and uses a memory-aware consistency loss to help learn a better joint representation of temporal context with random masked instructions.

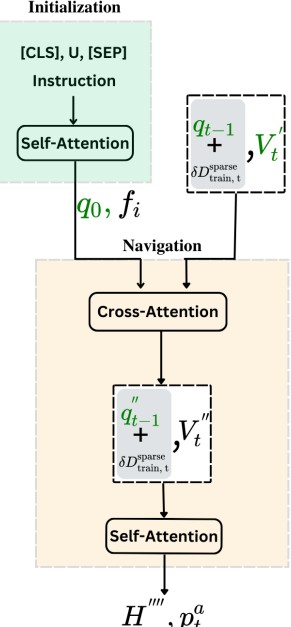

Figure 4: Reward token conditioning in VLN↺BERT-ORL. Initially, the language instruction is encoded by the self-attention module. During navigation, the sequence of state and visual tokens along with instruction features is passed multiple times through the cross-attention and self-attention modules to infer the action prediction probabilities.

Initially, the instruction is fed to the multi-layer transformer to get the language representation $X$. The convolutional vision encoder is then used to encode the image observations. At each step, the vision representation $V_t$ consists of the encodings of all the candidate image observations.

In order to learn cross-modality representations, a cross-modality encoder consisting of self-attention and cross-attention layers is used. Similar to VLN↺BERT, we add the reward token $\delta D_{\text{train, t}}^{\text{sparse}}$ to the [CLS] token embedding (X[0]) before we give it as input to the cross-modality encoder. The language representation X, vision representation $V_t$, and previous activations $M_t$ are then fed to the cross-modality encoder C as:

$$\hat{X}[0] = X[0] + \delta D_{\text{train, t}}^{\text{sparse}}$$

$$X = [\hat{X}[0]; X[1:]]$$

$$\hat{X}, \hat{M}_t, \hat{V}_t = C(X, [M_t; V_t])$$

where [;] denotes concatenation. The output $\hat{V}_t$ is then given as input to the action prediction head to make the action decision for this step:

$$a_t = MLP(\hat{V}_t)$$

The memory bank is updated at the end of each step by reusing the object activations $\hat{V}_t$ according to the current agent action decision as

$$M_t \leftarrow (M_{t-1}, [\hat{V}_t^k; d_t^k])$$

where k refers to the index of the selected vision output and $d_t^k$ is the corresponding directional feature of action at step t. A mixture of imitation learning loss and memory-aware consistency loss was used to train the network from scratch.

## B.2 Visualisations

Fig. 5 and 6 show the visualisation of the R2R trajectories of VLN↻BERT-ORL and VLN↻BERT trained on 30% noisy R2R dataset. Fig. 7 and 8 show the RxR trajectory visualisation for the VLN↻BERT-ORL and VLN↻BERT agents trained on 30% noisy RxR dataset. The panoramic view and the heading (view in the heading direction) are shown at each navigation step. In each trajectory, the reward-conditioned model correctly follows the instruction to reach the goal state whereas the baseline model gets distracted in between and doesn't reach the goal location. For example, in Fig. 5, the agent was supposed to go straight in the first navigation step but the VLN↻BERT agent took the other gate. Additionally, in Fig. 7, the VLN↻BERT agent initially got confused but was later able to exit the room. However, it kept walking and finally stopped at a wrong location.

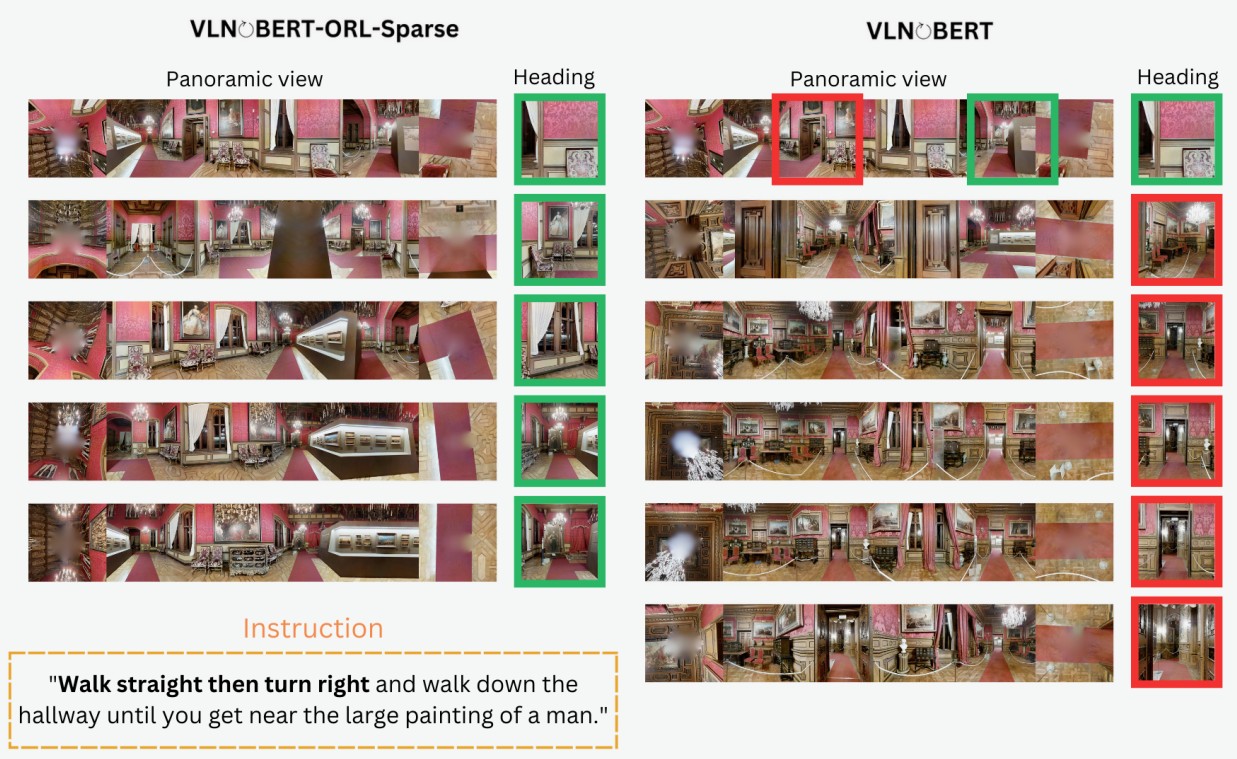

Figure 5: Visualisation of panoramic views and headings of VLN↻BERT-ORL and VLN↻BERT at every step in the trajectory.

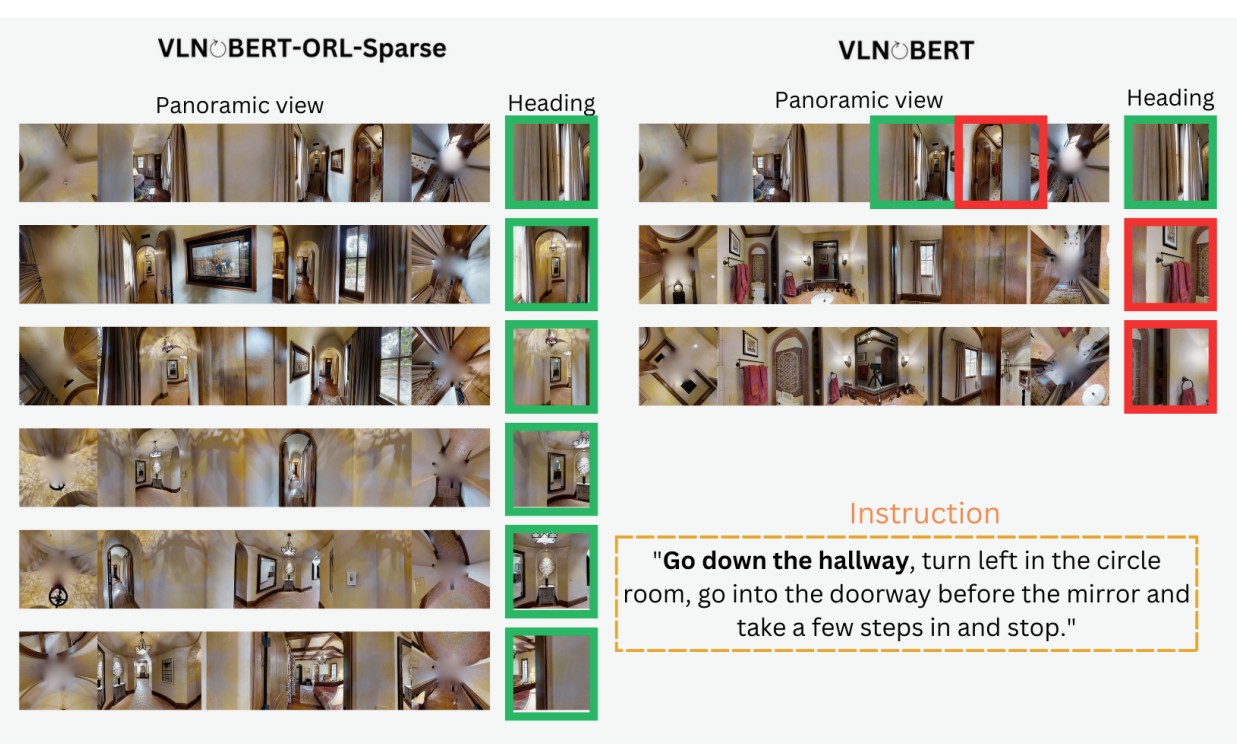

Figure 6: Visualisation of panoramic views and headings of VLN↻BERT-ORL and VLN↻BERT at every step in the trajectory.

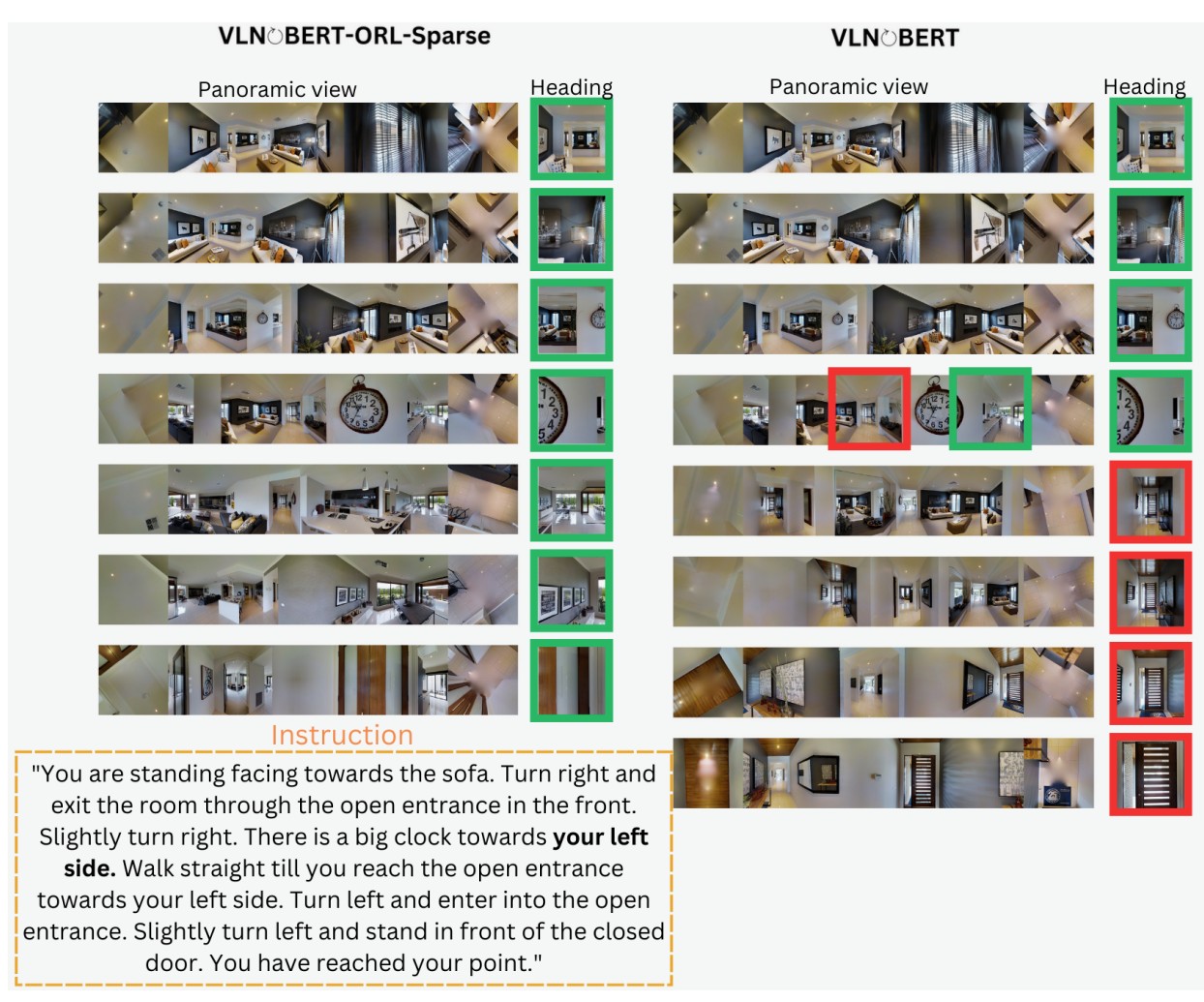

Figure 7: Visualisation of panoramic views and headings of VLN↻BERT-ORL and VLN↻BERT at every step in the trajectory.

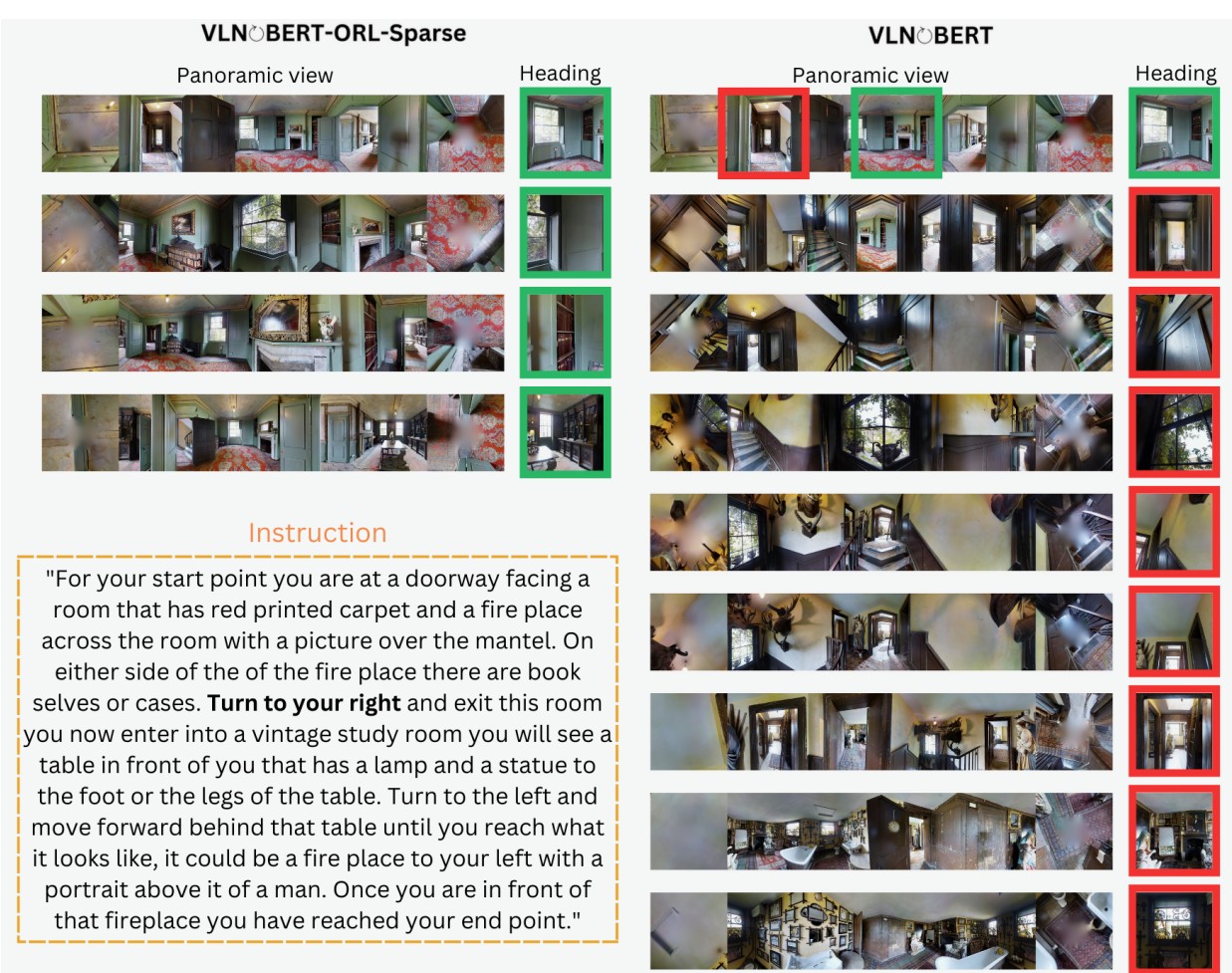

Figure 8: Visualisation of panoramic views and headings of VLN↻BERT-ORL and VLN↻BERT at every step in the trajectory.

## C   Additional related works on Offline RL

A common approach to offline RL is to use value-based methods where the goal is to learn a policy that maximizes a notion of expected cumulative reward, given the fixed dataset of interactions. However, they suffer from issues such as value overestimation and error propagation. Policy constraint methods Peng et al. (2019); Wu et al. (2019); Siegel et al. (2020); Kumar et al. (2019a) try to address this issue by constraining the learned policy to stay close to the behavioural policy. Regularization based methods penalize the large Q values to avoid issues pertaining to the OOD actions. Specifically, CQL Kumar et al. (2020) penalizes the large Q-values for state-action pairs not observed in the dataset. Implicit Q-learning Kostrikov et al. (2021) estimates the Q-values using a density ratio function and trains the policy network using importance-weighted regression.

