# OpenReview forum: "Scaling Vision-and-Language Navigation With Offline RL"
_TMLR — Accepted by TMLR_

### Review · Reviewer_vWx3 · 2024-01-18

**Summary Of Contributions:**

This paper studies reward-conditioned vision-and-language navigation problems, called VLN-ORL (vision-and-language navigation with offline RL). Previous vision-and-language navigation works often leverage expert demonstrations only, and employ data augmentation strategy or online exploration to tackle data shortage issue. By leveraging offline RL paradigm, the model could learn from sub-optimal demonstration data.

The proposed methods, VLN-BERT-ORL and MTVM-ORL, trains policies conditioned on reward tokens, and tests policies conditioned on  reward = +1. Such RL-based approaches achieve consistently better success rate (SR) and success rate weighted by path length (SRL), compared to Behavior Cloning (w/o any conditioning) baseline.

**Audience:**

Yes

**Broader Impact Concerns:**

I think broader Impact concerns are not applicable or are sufficiently addressed in this paper.

**Claims And Evidence:**

Yes

**Requested Changes:**

- Figure 1 is a bit confusing. I think step 3 in train data is shared among expert / suboptimal demos, but labeled with -1. Expert demonstrations also have sub-optimal steps?
- In addition, sub-optimal demonstrations finally reach goal (r=0), but they seem to be different from the goal in expert demonstration. Is it OK to be labeled with 0 even if the final destination is different from the oracle?
- In Table 5, some numbers are not bolded. I don't think this is intended.
- In Figure 3, RewardC VLN-BERT should be VLN-BERT-ORL for consistency? Moreover, it would be nice to add some short notes around each step (e.g. explaining why this step is OK/NG). The reader should infer the situation from those small pictures.

**Strengths And Weaknesses:**

### Strengths
- VLN-BERT-ORL and MTVM-ORL achieve consistently better SR and SRL than naive VLN-BERT and MTVM without reward token. The results are empirically strong.
- VLN-BERT-ORL and MTVM-ORL can learn from the suboptimal (noise-added, random) data, while naive VLN-BERT and MTVM face significant performance degradation. This might be a promising approach for vision-and-language navigation, since we do not always need to collect gold trajectories.

### Weaknesses
- In Table 1 & 5, naive VLN-BERT and MTVM sometimes achieves better navigation error than proposed methods, but I think the explanation / analysis are limited.
- It might be seen as naive adaptation of the reward-conditioned policies in offline RL, like Decision Transformer, to vision-and-language navigation domain.
- Offline RL requires positional information to compute reward. This means VLN-ORL can learn from sub-optimal data, but requires additional annotation about positions. There are some trade-offs. If positional information are available, VLN-BERT with position conditioning might be a possible baseline (I know this will face the issue in test time, but I'm not fully sure why positional information is available in training time, but unavailable in test time).

---

> ### Author Response · Authors · 2024-02-14
> **Author response to Reviewer vWx3 (Part 1)**
>
> We sincerely appreciate the thoughtful and constructive feedback you've provided for our paper. We have carefully considered your feedback and would like to address each of your points in detail.
>
> >In Table 1 & 5, naive VLN-BERT and MTVM sometimes achieves better navigation error than proposed methods, but I think the explanation / analysis are limited.
>
> We note that the reward-conditioned model occasionally demonstrates a higher navigation error than the baseline. This discrepancy can be attributed to our model being conditioned on a zero reward only when it reaches proximity to the goal. In cases where the agent fails to reach the goal, the conditioning of a +1 reward incentivizes continuous movement, leading to both higher navigation errors and increased trajectory lengths in certain instances. We appreciate your insightful feedback, and have incorporated this clarification into our paper to provide a more thorough understanding of the observations.
>
> >It might be seen as naive adaptation of the reward-conditioned policies in offline RL, like Decision Transformer, to vision-and-language navigation domain.
>
> We acknowledge the similarity between our approach and reward-conditioned policies in offline RL, such as Decision Transformer. However, our proposal introduces a novel reward token linked to the distance between consecutive states in the vision-and-language navigation domain.
> While Decision Transformer conditions the model on a returns-to-go token, indicative of the total reward for the remaining trajectory, our adaptation would associate this token with the agent's current distance from the goal. Although feasible during training, computing this returns-to-go token at test time presents challenges, necessitating real-time knowledge of the agent's distance from the goal at each timestep. In response, we propose a unique approach by conditioning the model on a novel reward token based on the distance between consecutive states. This design allows us to simplify the conditioning process, using rewards of +1/0 to guide the agent closer to the goal without the need for the total distance computation at every step. We believe this adaptation enhances the feasibility of applying reward-conditioned policies in the vision-and-language navigation domain.
>
> >Offline RL requires positional information to compute reward. This means VLN-ORL can learn from sub-optimal data, but requires additional annotation about positions. There are some trade-offs. If positional information are available, VLN-BERT with position conditioning might be a possible baseline (I know this will face the issue in test time, but I'm not fully sure why positional information is available in training time, but unavailable in test time).
>
> Indeed, in our approach, we leverage positional information during training to allow the model to learn from suboptimal actions. While this necessitates additional annotation about positions, we acknowledge the trade-offs involved. Our rationale for making positional information available during training is to facilitate learning from diverse suboptimal scenarios. However, once the model is trained, our aim is to achieve a level of autonomy where the agent doesn't rely on positional information, specifically the distance from the goal, at every step during execution. This design choice is made to enhance the practicality and real-world applicability of the model, minimizing the need for additional positional data during execution.
>
> >Requested Changes:
> >>Figure 1 is a bit confusing. I think step 3 in train data is shared among expert / suboptimal demos, but labeled with -1. Expert demonstrations also have sub-optimal steps?
>
> Thank you for your feedback regarding Figure 1. The top row illustrates an expert demonstration, while the second and third rows depict suboptimal demonstrations with negative reward values at some steps. We have also added this in the figure caption.
> So, step 3 has negative reward values at some steps as it corresponds to a suboptimal demonstration. Additionally, here we use Expert to refer to manually annotated data which differs from the Expert dataset we generate using HAMT. We have updated the figure to avoid this confusion. Expert demonstrations (manually annotated) don’t have suboptimal steps whereas the Expert dataset generated with HAMT might have suboptimal steps as HAMT might take wrong actions at certain steps.

---

> ### Author Response · Authors · 2024-02-14
> **Author response to Reviewer vWx3 (Part 2)**
>
> >>In addition, sub-optimal demonstrations finally reach goal (r=0), but they seem to be different from the goal in expert demonstration. Is it OK to be labeled with 0 even if the final destination is different from the oracle?
>
> The numerical value below each image represents the reward used for conditioning the model at that step. In this context, a positive reward signifies that the agent is intended to move towards the goal at that step, while a negative reward indicates the agent should move away from the goal. When the model is conditioned on a zero reward, it implies that the agent is expected to halt its movement whether it has reached the goal or not. In the figure in both the suboptimal demonstrations the agent doesn’t reach the goal at the end but is conditioned on a zero reward so as to make it stop at that step.
>
> >> In Table 5, some numbers are not bolded. I don't think this is intended.
>
> Thank you for bringing this to our attention. We apologize for any oversight in Table 5. We have appropriately bolded the necessary numbers in the updated version.
>
> >> In Figure 3, RewardC VLN-BERT should be VLN-BERT-ORL for consistency? Moreover, it would be nice to add some short notes around each step (e.g. explaining why this step is OK/NG). The reader should infer the situation from those small pictures.
>
> We acknowledge the error regarding the label "RewardC VLN-BERT" in Figure 3, which should indeed be "VLN-BERT-ORL" for consistency. We apologize for any confusion this may have caused and will ensure that the correction is made accordingly. Additionally, we have made some updates to enhance the overall readability of the figure. We have also made the caption more detailed so as to explain what the agent does at each step.
>
> We have submitted a revised version of the manuscript where all revisions are highlighted in blue. We are thankful to you for your valuable inputs, which have contributed to improving the quality of our work.

---

### Review · Reviewer_mjiP · 2024-01-22

**Summary Of Contributions:**

1. The paper proposed to study the problem for learning from noisy data (without active exploration). This is a realistic consideration as (a) the real data captured from world can be noisy. (b) active exploration by agents can lead to destructive affects.
2. The paper built up a dataset and evaluation protocol to study the effectiveness of this method.
3. The paper used an offline reinforcement learning algorithm to solve the proposed problem. The offline RL takes the expected local reward as a condition. This reward is available during training and set to be constant during inference.
4. The paper shows improvement over the proposed experimental setup and provide comprehensive analysis.

**Audience:**

Yes

**Broader Impact Concerns:**

n.a.

**Claims And Evidence:**

Yes

**Requested Changes:**

Question:
1. The paper claims that it would like to solve the questions "How can we achieve effective data scaling without resorting to online exploration, a strategy that could pose safety concerns?". However, it is not clear to me why the proposed method is free of this risky exploration. Since the offline RL needs some non-perfect routes for the training and it can introduce unsafe actions. Thus I would like a paragraph which precisely clarify the difference of the approches (previous RL and the offline RL) in the paper.
2. The paper assumes "training trajectories corresponds to a single pair of 2D goal coordinates" because of the "instructions" are ambiguous. Given that, is it better to consider some more concrete tasks such as "image-goal navigation"? It can intrinsically resolve this concern.
3. Please saw the weakness as well.

Issues:
1. It seems that "episode horizon" in Sec 3.Setup is not a well-defined terminology. The meaning is not clear to me and also it's hard to find on Internet.

**Strengths And Weaknesses:**

**Strengths**
1. Please see the above contribution list. In general, I like this novel problem and the efforts behind to make it trackable.

**Weakness**
1. The concern regarding the high-quality data scarcity is valid to me. However, the simulator indoor-room navigation might not be the optimal setup to study it. Since we actually can construct good navigation data under this setup, and we are not sure about whether the "
simulated" imperfect data reflects the actual noise (as listed in the intro of the paper). The presentation of the method can be better with a real inaccurate data setup and also with the proposed method.
2. The design of the reward highly depends on well topology of the. It generally assumes that the graph of the environments has strong connectivity. For sparse-connected network, e.g.,
    ```
   A1 -- A2 --- A3 .... -- An
                           |
   B1 -- B2 --- B3 .... -- Bn
     ```
   drawn in geometric distance. Most of the paths (e.g., from A2 --> B1) would involve a long path of negative reward -1 (i.e., from A2 to An). I think that the current reward design is not robust to such sparse graph. The illustration of the above happens in real life as well, e.g., for building with two floors where the ladders are only available at corners. Under this env, the navigation goal across floors can possibly trigger the long-path negative rewards.
3. The dataset is constructed from a pre-trained VLN model HAMT. I have a little bit trouble in persuading myself regarding the effectiveness of such dataset. The subpoints are: (a) as stated in the paper, the actions are predicted with model that has been trained on the given dataset; if the model is capable enough, the generated paths should reveal the training path? (b) The noise takes random actions; would it be better to still sample from the policy but with higher temperature (i.e., a smoothed distribution)? (c) What's the accuracy of the HAMT model under `Expert', `15%-noisy data', and `30%-noisy data'? I would like to understand how "noisy" these data are given whether the final state of the paths reach the target goal.
4. I also got a little bit confused on interpreting the results. Since the training path are generated by the HAMT model, this strategy is similar to a distillation approach. Thus, the confusion to me is whether the HAMT-generated data is actually noisy, instead of some reasonable data can be helpful to training (as distillation).


Overall, I think that it is a new problem formulation to me and I would like to see these work come to the community. However, I would like to convince myself that the experimental setup is somewhat reasonable and the improvement over this experimental setup (by following work) can faithfully reflect the research progress.

---

> ### Author Response · Authors · 2024-02-15
> **Author response to Reviewer mjiP (Part 1)**
>
> Thank you for your thoughtful assessment of our work and for providing insightful comments and suggestions. We value your feedback and would like to address your points to offer a more comprehensive explanation of our approach.
>
> >Weakness
> >>1. The concern regarding the high-quality data scarcity is valid to me. However, the simulator indoor-room navigation might not be the optimal setup to study it. Since we actually can construct good navigation data under this setup, and we are not sure about whether the " simulated" imperfect data reflects the actual noise (as listed in the intro of the paper). The presentation of the method can be better with a real inaccurate data setup and also with the proposed method.
>
> We acknowledge the significance of assessing our proposed method within contexts that closely resemble real-world scenarios. Our dataset generation process incorporates deliberate noise by enabling the HAMT model to take random actions at various steps. This deliberate injection of randomness aims to capture the uncertainties and imperfections akin to those encountered in real-world scenarios. Importantly, we have experimented with different probabilities of the model taking random actions, ranging from 0 to 100%. This nuanced exploration allowed us to generate suboptimal datasets of varying difficulties, providing a comprehensive evaluation framework. Furthermore, the inclusion of the Random dataset, characterized as a completely noisy dataset, represents an extreme scenario to test the robustness of our proposed method under challenging conditions.
>
> Nevertheless, it's essential to acknowledge the inherent limitations of simulations, as they may not fully encapsulate all the intricacies of real-world data. Despite these constraints, we believe that our deliberate injection of noise through random actions offers an effective representation of actual noise scenarios. This approach allows us to systematically study the impact of suboptimal data on our proposed method in a controlled and measurable environment. We hope this clarification underscores the relevance and effectiveness of our dataset generation method in addressing the concerns raised.
>
> >>2. The design of the reward highly depends on well topology of the. It generally assumes that the graph of the environments has strong connectivity. For sparse-connected network...
>
> To assess the robustness of the current reward design in the context of sparse graphs, we conducted a comprehensive analysis on trajectories within the RxR validation sets. As discussed in the paper, we had previously explored the performance enhancement resulting from reward-conditioning on trajectories involving at least one deviation from the goal (we refer to the set of such trajectories as T1 and number of such trajectories as N1).
> To provide a more nuanced understanding, we extended our analysis to explore trajectories where the agent consecutively deviates from the goal at least twice, denoted as N2. Additionally, we examined cases of consecutive deviations denoted as N3, N4, and N5, where Ni represents the number of trajectories where the agent moves away from the goal consecutively at least i times. We discuss the performance improvements on those sets in the “Is it really not too greedy” subsection of Ablation Studies in the updated paper.
>
>
> >>3. The dataset is constructed from a pre-trained VLN model HAMT. I have a little bit trouble in persuading myself regarding the effectiveness of such dataset. The subpoints are: (a) as stated in the paper, the actions are predicted with model that has been trained on the given dataset; if the model is capable enough, the generated paths should reveal the training path? (b) The noise takes random actions; would it be better to still sample from the policy but with higher temperature (i.e., a smoothed distribution)? (c) What's the accuracy of the HAMT model under Expert', 15%-noisy data', and `30%-noisy data'? I would like to understand how "noisy" these data are given whether the final state of the paths reach the target goal.
>
> While the HAMT model has indeed been trained on the VLN dataset, our deliberate injection of noise ensures that the true paths are not revealed. To assess the noise level, we checked how many trajectories in each dataset ended with the agent not reaching the goal state. The column "Fail" indicates the number of trajectories where the agent did not reach the goal.
> Here are the results:
>
> For R2R generated datasets:
> | Dataset | Fail | Total trajectories |
> | ----- | ---- | ----- |
> | Expert| 1653 | 14039 |
> | 15%   | 5115 | 14039 |
> | 30%   | 7944 | 14039 |
> | Random| 13696 | 14039 |
>
> For RxR generated datasets:
>  Dataset|  Fail        | Total trajectories  |
> | -----     | ----          | -----                      |
> | Expert| 8709       | 26464                  |
> | 15%   | 14749     | 26464                  |
> | 30%   | 18451     | 26464                  |
> | Random| 24841 | 26464                   |

---

> ### Author Response · Authors · 2024-02-15
> **Author response to Reviewer mjiP (Part 2)**
>
> As is evident from the analysis, a significant number of trajectories indicate instances where the agent did not reach the goal in each dataset. Moreover, the count of such trajectories tends to increase with the rise in noise within the dataset. Sampling from the HAMT policy but with higher temperature would lead to a diverse set of samples as the model is encouraged to explore a wider range of possibilities rather than selecting the most likely action and could be an alternative way of generating suboptimal datasets.
> >>3. I also got a little bit confused on interpreting the results. Since the training path are generated by the HAMT model, this strategy is similar to a distillation approach. Thus, the confusion to me is whether the HAMT-generated data is actually noisy, instead of some reasonable data can be helpful to training (as distillation).
>
> As demonstrated in our earlier analysis, the HAMT-generated data indeed exhibits a notable degree of noise, and this noise level varies across different datasets generated. Specifically, training on the Expert dataset aligns more closely with a distillation approach, where the training paths are inferred from the HAMT model. However, for the other datasets, we intentionally introduce varying levels of noise during data generation. This deliberate inclusion of noise distinguishes the training strategy from a conventional distillation approach. Our goal in incorporating noise is to assess the robustness and adaptability of our model to less ideal or noisy scenarios.
>
> >Requested Changes:
> >>1. The paper claims that it would like to solve the questions "How can we achieve effective data scaling without resorting to online exploration, a strategy that could pose safety concerns?". However, it is not clear to me why the proposed method is free of this risky exploration. Since the offline RL needs some non-perfect routes for the training and it can introduce unsafe actions. Thus I would like a paragraph which precisely clarify the difference of the approches (previous RL and the offline RL) in the paper.
>
> Thank you for your insightful comment. It's essential to clarify the distinction between imperfect and unsafe demonstrations in the context of offline RL. While imperfect demonstrations may not follow optimal routes, not all of them lead to unsafe actions. The offline RL setting assumes the existence of a logged dataset of imperfect demonstrations. Notably, this dataset is not collected by the learning agent itself; instead, it is assumed to be collected by other agents or behavioral policies. In the paper, we also discuss real-world scenarios where such data could be obtained, including human navigation data in congested urban settings and AI agents navigating imperfect simulated environments, such as a dynamic shopping mall.
>
> The learning agent under consideration is not actively collecting the offline data and, by extension, not engaging in any unsafe actions during this process. More generally, our main contribution in this work is not related to questioning how these imperfect demonstrations would be collected; it is more related to the question of how one can learn a policy from offline suboptimal datasets. This aligns with the common setup in prior works on offline RL[1,2]. We have also added a paragraph for this in the paper to ensure a more precise understanding.
>
> Related works:
> 1. Fu, Justin, Aviral Kumar, Ofir Nachum, George Tucker, and Sergey Levine. "D4rl: Datasets for deep data-driven reinforcement learning." arXiv preprint arXiv:2004.07219 (2020).
> 2. Fujimoto, S., Conti, E., Ghavamzadeh, M., & Pineau, J. (2019). Benchmarking batch deep reinforcement learning algorithms. arXiv preprint arXiv:1910.01708.
>
> >>2. The paper assumes "training trajectories corresponds to a single pair of 2D goal coordinates" because of the "instructions" are ambiguous. Given that, is it better to consider some more concrete tasks such as "image-goal navigation"? It can intrinsically resolve this concern.
>
> Thank you for your insightful comment regarding the consideration of concrete tasks such as "image-goal navigation". While we agree that having image-goal information along with language instructions would enhance the comprehensibility of the training trajectories, it's essential to highlight the broader context. In certain scenarios, we may only have access to the language instruction and not the corresponding image of the goal. Our primary objective was to develop a method capable of functioning even when only the language instruction is available. That being said, we fully recognize the potential benefits of extending our approach to incorporate image-goal information if it is present. Our method is designed to be flexible, and we believe it could be extended to leverage both language and image cues when available.

---

> > ### Author Response · Authors · 2024-02-15
> > **Author response to Reviewer mjiP (Part 3)**
> >
> > >Issues:
> > >>It seems that "episode horizon" in Sec 3.Setup is not a well-defined terminology. The meaning is not clear to me and also it's hard to find on Internet.
> >
> > In the context of vision-based navigation setups, the term "episode horizon" refers to the predefined maximum duration or length of an episode during which an agent interacts with its environment to achieve a specific goal. It serves as a control mechanism, defining the temporal scope within which the agent must complete its navigation task. We have updated this in the paper as well.
> >
> > We have submitted a revised version of the manuscript, where the revisions are highlighted in blue. We greatly appreciate your insightful feedback, which have helped us in improving the quality of our work.

---

### Review · Reviewer_5e26 · 2024-02-05

**Summary Of Contributions:**

The authors present VLN-ORL, a simple reward-conditioning objective for VLN that adds a dense L2-distance based reward signal to condition models on so that they can learn from suboptimal data. They also introduce a new dataset/benchmark to evaluate VLN agents in simulation with suboptimal datasets.

**Audience:**

Yes

**Claims And Evidence:**

No

**Requested Changes:**

Unfortunately, while the paper is well-written and interesting, I am not convinced that the reward-conditioning is more effective than existing methods due to the greediness of the reward conditioning. I would like to see every experimental concern addressed before acceptance, namely a comparison with return-conditioned offline RL (with the same models), regular offline RL methods (e.g., with a Q-transformer inspired model), and an ablation study which conclusively demonstrates that the reward mechanism being greedy is fine.

This reward-conditioning isn’t really a novel idea either, but I think that is OK given they show it works in VLN specifically and that the reward requires L2 distance, which is specific to navigation tasks.

Every other concern is minor and would strengthen the work if improved upon.

**Strengths And Weaknesses:**

### Strengths:

**Writing:** Overall this is a well-written, clear, and easy to understand paper.

**Dataset:** The authors craft a new dataset/benchmark for their VLN which includes suboptimal trajectories. I think this is a valuable contribution.

**Experiments:** Ablation studies are comprehensive for the most part.

### **Weaknesses:**

**Clarity:**

- Figure 1 isn’t very attractive, and more importantly, isn’t that helpful currently:
    - It’s very hard to realize what’s going on by looking at the figure alone, and then if I read the caption by itself, it’s a simple enough description to not need the figure at all. I would recommend trying to make the figure itself more clear and make a larger distinction between trajectories and images, what the rewards are doing, arrows pointing out what comes from the what dataset, replacing text with shapes (e.g.: expert demos being shaped as a square is not informative, if it’s shaped as a dataset icon then it’s easy to understand what it is without *needing* to read the label. And there’s no distinction between “expert demos” square and the “VLN-ORL” square but one is referring to a dataset and another is referring to your method.).
    - It may also be good to simply show how the reward token is used in a generic (transformer) model architecture in this figure next to VLN-ORL, rather than just labeling the rewarded trajectories with VLN-ORL

**Claims/Framing:**

- “However, we focus on developing techniques that can learn from offline suboptimal datasets without the need for exploration..” I would tone this statement down quite a bit, it’s debatable if there will ever be methods which do not require *any* exploration to do unseen VLN, especially given that this paper is only evaluated in simulation and still getting ~50% success rates. I believe that VLN-ORL can actually be complementary to the related works this statement is in relation to, so it should just be framed that way.

**Method Novelty/Technical Contribution:**

- The method is to add a reward conditioning token, just like Decision Transformers and other return-conditioned RL approaches, but which is densified by using reward *differences.* I think this kind of reward densifying is usually treated as an engineering trick and it is definitely not a novel one. I’m not sure of any major technical contribution here, especially given the experiments do not compare against any other offline RL approaches to demonstrate that this specific reward-conditioning works better.
- In addition, this reward-conditioning has clear problems that actual, non-conditioned offline RL wouldn’t have. If you force the model to be conditioned on reward differences and always condition it on a positive reward difference at test time, then it will be difficult for it to deviate from this behavior. Return-conditioning solves this problem, but the authors noted it’s hard to predict returns at test time. This still warrants a fair comparison, as return-conditioned offline RL has been shown to have some success even when the test-time return is unknown.
    - Even if you don’t condition on returns, regular Offline RL will still be able to stitch together trajectories and overcome suboptimal actions within a single trajectory,. I don’t see any evidence from the proposed method or the experiments that VLN-ORL can avoid this issue.

**Experiments**

- Due to the above weaknesses in the technical contributions, I think comparing against return-conditioning is required for acceptance.
- In addition, there should be comparisons against regular offline RL, because the authors are proposing an offline RL method. You could modify the existing VLN architectures with Q-transformer: [Q-Transformer (qtransformer.github.io)](https://qtransformer.github.io/), or follow this for implementing IQL on transformers [[2206.11871] Offline RL for Natural Language Generation with Implicit Language Q Learning (arxiv.org)](https://arxiv.org/abs/2206.11871), for example.
- Ablation study: The ablation study claims to show that the greedy reward maximization objective prevents the agent from getting stuck. But inclusion criteria for the “val-tough” dataset is just a minimum of one action deviation from the optimal goal trajectory. This doesn’t conclusively demonstrate much, because a deviation of 1 action from the optimal goal trajectory is likely easy to overcome, when actions are learned and averaged over a large dataset, even with a greedy objective.
    - To prove this is actually true, it authors should partition “val-tough” into multiple subsets of various numbers of action deviations, and compare performance improvement over baselines at each # of action deviations.
- Standard deviation: I know it’s not that common to have standard deviations reported in these supervised learning settings, but given Table 3 showing varying levels of action prediction accuracy deviation based on small dataset differences, the authors should report numbers averaged over multiple model training seeds too.

**Minor Issues:**

- Page 5: wrong closing quotation markers for “go to the room”
- Bottom of methods section: Appendix A link doesn’t work
- Table 3 is too small

---

> ### Author Response · Authors · 2024-02-18
> **Author response to Reviewer 5e26 (Part 1)**
>
> We are grateful for the thorough evaluation of our work and the insightful comments and suggestions you have provided. We have carefully considered each point raised and would like to provide detailed responses to address them effectively.
>
> >Clarity: Figure 1 isn’t very attractive, and more importantly, isn’t that helpful currently:
>
> Thank you for your thorough feedback. We've enhanced clarity by using dataset icons and arrows to illustrate the trajectory sources. Furthermore, we've made efforts to visually separate rewards and images to improve overall clarity and understanding. The caption has been updated for better understanding of the figure's content. We hope these adjustments address your concerns and make the figure more informative and visually appealing.
>
> >Claims/Framing: “However, we focus on developing techniques that can learn from offline suboptimal datasets without the need for exploration..” I would tone this statement down quite a bit, it’s debatable if there will ever be methods which do not require any exploration to do unseen VLN, especially given that this paper is only evaluated in simulation and still getting ~50% success rates. I believe that VLN-ORL can actually be complementary to the related works this statement is in relation to, so it should just be framed that way.
>
> Thank you for your insightful feedback. We've revised the statement to better align with the context and acknowledge the ongoing challenges in VLN. This is the revised statement: “However, we focus on developing techniques that can learn from offline sub-optimal datasets reducing the need for exploration and developing manually annotated datasets.”. By framing our focus as developing techniques that reduce the need for online exploration, rather than eliminating it entirely, we aim to provide a more balanced perspective. We appreciate your input and believe that this adjustment better reflects the contribution and potential of our approach.
>
> >Method Novelty/Technical Contribution:
> >>1. The method is to add a reward conditioning token, just like Decision Transformers and other return-conditioned RL approaches, but which is densified by using reward differences. I think this kind of reward densifying is usually treated as an engineering trick and it is definitely not a novel one. I’m not sure of any major technical contribution here, especially given the experiments do not compare against any other offline RL approaches to demonstrate that this specific reward-conditioning works better.
>
> We respectfully object to the positioning of our work as an "engineering trick" and "definitely not a novel one". Firstly, there is ample evidence that engineering is fundamental to AI development. Secondly, we would greatly appreciate it if the reviewer could provide specific insights into why they perceive our approach as being not novel. If there are concrete references which have used the same conditioning strategy, we will be happy to cite and discuss them.
>
> Furthermore, we recognize the importance of comparing our method against other offline RL approaches to demonstrate its effectiveness comprehensively. We have incorporated comparisons with the return-conditioned model in our paper, illustrating the superior performance of our specific reward conditioning approach.
>
> >>2. In addition, this reward-conditioning has clear problems that actual, non-conditioned offline RL wouldn’t have. If you force the model to be conditioned on reward differences and always condition it on a positive reward difference at test time, then it will be difficult for it to deviate from this behavior. Return-conditioning solves this problem, but the authors noted it’s hard to predict returns at test time. This still warrants a fair comparison, as return-conditioned offline RL has been shown to have some success even when the test-time return is unknown. Even if you don’t condition on returns, regular Offline RL will still be able to stitch together trajectories and overcome suboptimal actions within a single trajectory,. I don’t see any evidence from the proposed method or the experiments that VLN-ORL can avoid this issue.
>
> Although, it might seem that the reward token used for conditioning the model is too greedy and can lead the agents to get stuck in cases where they need to consider the long-term impact of their actions, we show through additional experiments that reward-conditioning leads to performance enhancements even in scenarios involving consecutive deviations from the goal. Further details and analysis can be found in the "Is it really too greedy" subsection of our Ablation Studies.

---

> > ### Author Response · Authors · 2024-02-18
> > **Author response to Reviewer 5e26 (Part 2)**
> >
> > >Experiments
> > >>1. Due to the above weaknesses in the technical contributions, I think comparing against return-conditioning is required for acceptance.
> >
> > Thank you for your valuable inputs. We have taken your suggestion into account and conducted a comparison against return-conditioning. In Table 1, we present the results of this comparison. The conditioning mechanism remains the same, involving addition of the returns-to-go token to the state token at multiple steps in the pipeline. During training, at each timestep, we define the returns-to-go token as the distance of the agent from the goal at that particular step. At test time, since we cannot assume knowledge of the agent's distance from the goal at each step, we initialize the returns-to-go token with the maximum length of trajectories in the validation set. Subsequently, we continuously update it by subtracting the distance traveled by the agent at each step. To ensure a fair comparison, we set the returns-to-go token to zero when the agent reaches close to the goal, under the assumption that the agent can detect its proximity to the goal. We report all the results with the test-time initial return as the maximum length of trajectories in the validation set. However, we also include a performance comparison using the average length of trajectories in the validation set as the initial return in Table 8.
> >
> > Our findings reveal that reward-conditioning significantly outperforms return-conditioning across all proposed datasets. Interestingly, while the return-conditioned model outperforms the baseline (VLNBERT) on most of the R2R datasets, it demonstrates less effectiveness on the RxR datasets. This discrepancy may be attributed to the significant variance in trajectory lengths within the RxR dataset, which hinders generalization when initializing the test-time returns based on the maximum trajectory length. Consequently, this results in a decline in performance, highlighting the limitations of return-conditioning in scenarios where test-time returns are unknown. Furthermore, as previously discussed, we present the performance comparison of return-conditioned VLNBERT on RxR datasets with different test-time initial returns in Table 8. The results illustrate that using the maximum length of trajectories as the initial return yields superior performance compared to using the average length.
> >
> > We observe that return-conditioning struggles with the high variance in trajectory lengths within the RxR validation sets, as it requires setting an initial return value. In contrast, reward-conditioning does not rely on distance from the goal at test time, avoiding this issue.
> >
> > >>2. In addition, there should be comparisons against regular offline RL, because the authors are proposing an offline RL method. You could modify the existing VLN architectures with Q-transformer: Q-Transformer (qtransformer.github.io), or follow this for implementing IQL on transformers [2206.11871] Offline RL for Natural Language Generation with Implicit Language Q Learning (arxiv.org), for example.
> >
> > We experimented with modifying VLNBERT with Q-transformer; however, we encountered significant challenges with unstable training and low performance. Here are the initial results on some of the RxR datasets:
> >
> > | Dataset   | Val seen |       |      |      |
> > |:---------:|:--------:|:-----:|:----:|:----:|
> > |           | TL       | NE    | SR   | SPL  |
> > | Expert    | 6.7      | 12.62 | 8.57 | 7.74 |
> > | 15% Noisy | 9        | 13.18 | 9.39 | 8.71 |
> > | 30% Noisy | 10.39    | 13.04 | 9.36 | 8.27 |
> >
> >
> > | Dataset   | Val unseen |       |       |      |
> > |:---------:|:----------:|:-----:|:-----:|:----:|
> > |           | TL         | NE    | SR    | SPL  |
> > | Expert    | 6.26       | 11.91 | 9.49  | 8.55 |
> > | 15% Noisy | 8.51       | 12.63 | 9.91  | 8.76 |
> > | 30% Noisy | 10.09      | 12.43 | 10.09 | 8.44 |
> >
> > The tables clearly illustrates the poor performance of the Q-transformer-based VLNBERT model across all datasets. It seems that achieving satisfactory results with this approach would require significant engineering efforts, which could be pursued by future works. Nevertheless, our primary focus and scope lie in the realm of return-conditioned offline RL. We particularly work on conditioning based approaches as they can be seamlessly aligned with the established VLN architectures and objectives in use currently.

---

> > > ### Author Response · Authors · 2024-02-18
> > > **Author response to Reviewer 5e26 (Part 3)**
> > >
> > > >>3. Ablation study: The ablation study claims to show that the greedy reward maximization objective prevents the agent from getting stuck. But inclusion criteria for the “val-tough” dataset is just a minimum of one action deviation from the optimal goal trajectory. This doesn’t conclusively demonstrate much, because a deviation of 1 action from the optimal goal trajectory is likely easy to overcome, when actions are learned and averaged over a large dataset, even with a greedy objective.
> > > To prove this is actually true, it authors should partition “val-tough” into multiple subsets of various numbers of action deviations, and compare performance improvement over baselines at each # of action deviations.
> > >
> > > We've addressed your suggestion by partitioning the "val-tough" sets into subsets with varying numbers of action deviations, ranging from 1 to 5. Subsequently, we evaluated the proposed method, the baseline, and the return-conditioned approach on each of these subsets. More details about these experiments can be found in the "Is it really not too greedy" subsection of the Ablation Studies section.
> > >
> > > >>4. Standard deviation: I know it’s not that common to have standard deviations reported in these supervised learning settings, but given Table 3 showing varying levels of action prediction accuracy deviation based on small dataset differences, the authors should report numbers averaged over multiple model training seeds too.
> > >
> > > We have performed experiments using three different model seeds, and the corresponding outcomes are now presented in Table 4. Specifically, we display the results for three different subsets of data: 25%, 50%, and 75%. In each instance, we maintain the data subset fixed and train the model using three different seeds on that particular dataset. Notably, the low standard deviations suggest that the model’s performance exhibits minimal variation with changes in the random seed, highlighting its robustness and stability across different initial conditions.
> > >
> > > >Minor Issues:
> > > >>1. Page 5: wrong closing quotation markers for “go to the room”
> > > >>2. Bottom of methods section: Appendix A link doesn’t work
> > > >>3. Table 3 is too small
> > >
> > > Thanks for your detailed feedback. We have made these changes in the updated version of the paper.
> > >
> > > >Requested Changes:
> > > >>Unfortunately, while the paper is well-written and interesting, I am not convinced that the reward-conditioning is more effective than existing methods due to the greediness of the reward conditioning. I would like to see every experimental concern addressed before acceptance, namely a comparison with return-conditioned offline RL (with the same models), regular offline RL methods (e.g., with a Q-transformer inspired model), and an ablation study which conclusively demonstrates that the reward mechanism being greedy is fine.
> > > This reward-conditioning isn’t really a novel idea either, but I think that is OK given they show it works in VLN specifically and that the reward requires L2 distance, which is specific to navigation tasks.
> > > Every other concern is minor and would strengthen the work if improved upon.
> > >
> > > Thank you for your comprehensive assessment of our work and for recognizing its clarity and significance. In this work, we propose a novel reward token which allows flexible conditioning of VLN agents during training and testing. We agree that direct comparisons with existing methods are crucial for a comprehensive understanding of the proposed approach's efficacy. To address this, we have incorporated comparisons with return-conditioned VLNBERT, highlighting the efficacy of reward conditioning, particularly in navigation tasks. Also, it seems that adapting regular offline RL based methods to VLN would require substantial engineering efforts which we believe could be pursued by future works. Furthermore, despite the inherent greediness of our proposed reward token, our experiments consistently showcase its significant performance enhancements, even in challenging scenarios. Therefore, we believe it is reasonable to anticipate that the efficacy of the proposed greedy reward conditioning technique would persist even in intricate and complex environments.
> > >
> > > A revised version of our manuscript, incorporating the suggested revisions highlighted in blue, has been submitted. We sincerely appreciate your insightful feedback, which has been instrumental in enhancing the quality of our work.

---

> ### Comment · Reviewer_5e26 · 2024-02-18
> **Response to author rebuttal**
>
> Thanks for the detailed rebuttal and updating the paper.
>
> I have read through the entire rebuttal and the new draft, and here is a detailed response to certain things:
>
>
> > We respectfully object to the positioning of our work as an "engineering trick" and "definitely not a novel one". Firstly, there is ample evidence that engineering is fundamental to AI development. Secondly, we would greatly appreciate it if the reviewer could provide specific insights into why they perceive our approach as being not novel. If there are concrete references which have used the same conditioning strategy, we will be happy to cite and discuss them.
>
> First of all, I think the comparison against the offline RL baselines I proposed is great and strongly impacts the paper's ability to sell this claim. And I agree that "engineering tricks" are fundamental to AI development.
>
> Looking back at the original review, I believe the reason I had an issue with the novelty of the reward token is really that the proposed "reward token" consists of two components: 1) conditioning on a token representing the reward, which goes back to at least 2019 (https://arxiv.org/abs/1912.13465, "RCP-A" advantage conditioned variant is essentially reward-conditoning), and 2) doing reward shaping using information from the future state, which goes back to at least 1999 (https://people.eecs.berkeley.edu/~russell/papers/icml99-shaping.pdf, which introduces the concept of "reward potentials" for reward shaping as some general function $\phi(s') - \phi(s)$ which the proposed reward conditioning token definitely is a subset of.). The actual form of the reward, Euclidean distance, is also of course not novel and has been used as a reward function in countless papers.
>
> The best way to address this might be to give some context and citations to the reward-shaping literature (which I'm admittedly not extremely familiar with) when introducing the method itself on Page 6, and probably also contrast with the linked reward-conditioned policies paper from 2019 somewhere in the intro, page 6, or related works.
>
> In context of the updated results, I do believe this paper has merit, but I think this is an important point to resolve as readers familiar with offline RL and reward-conditioned policies might be confused about the novelty or at least the relation to prior work when reading this paper.
>
>
> > Our findings reveal that reward-conditioning significantly outperforms return-conditioning across all proposed datasets. Interestingly, while the return-conditioned model outperforms the baseline (VLNBERT) on most of the R2R datasets, it demonstrates less effectiveness on the RxR datasets. This discrepancy may be attributed to the significant variance in trajectory lengths within the RxR dataset, which hinders generalization when initializing the test-time returns based on the maximum trajectory length. Consequently, this results in a decline in performance, highlighting the limitations of return-conditioning in scenarios where test-time returns are unknown. Furthermore, as previously discussed, we present the performance comparison of return-conditioned VLNBERT on RxR datasets with different test-time initial returns in Table 8. The results illustrate that using the maximum length of trajectories as the initial return yields superior performance compared to using the average length.
>
> I appreciate the new experiments, but I studied the baseline and I think it's more fair to initialize the returns-to-go token with something **less** than the maximum trajectory length at each timestep. By initializing it with the max, you are literally asking the model for poorer performance on at least half the trajectories assuming the max length isn't the same for all trajs. But in the decision transformer paper, they initialize test-time returns optimistically, not pessimistically, for best perofrmance.
> A more fair value could just be the average instead. Can the authors perform this simple comparison too?
>
> > A revised version of our manuscript, incorporating the suggested revisions highlighted in blue, has been submitted. We sincerely appreciate your insightful feedback, which has been instrumental in enhancing the quality of our work.
>
> Regarding all other changes, I think they are great and have changed my overall opinion of the paper.
> I will recommend acceptance after the above changes are made. Thank you!

---

> ### Author Response · Authors · 2024-02-18
> **Author response to Reviewer 5e26**
>
> >First of all, I think the comparison against the offline RL baselines I proposed is great and strongly impacts the paper's ability to sell this claim. And I agree that "engineering tricks" are fundamental to AI development. Looking back at the original review, I believe the reason I had an issue with the novelty of the reward token is really that the proposed "reward token" consists of two components: 1) conditioning on a token representing the reward, which goes back to at least 2019 (https://arxiv.org/abs/1912.13465, "RCP-A" advantage conditioned variant is essentially reward-conditoning), and 2) doing reward shaping using information from the future state, which goes back to at least 1999 (https://people.eecs.berkeley.edu/~russell/papers/icml99-shaping.pdf, which introduces the concept of "reward potentials" for reward shaping as some general function which the proposed reward conditioning token definitely is a subset of.). The actual form of the reward, Euclidean distance, is also of course not novel and has been used as a reward function in countless papers. The best way to address this might be to give some context and citations to the reward-shaping literature (which I'm admittedly not extremely familiar with) when introducing the method itself on Page 6, and probably also contrast with the linked reward-conditioned policies paper from 2019 somewhere in the intro, page 6, or related works. In context of the updated results, I do believe this paper has merit, but I think this is an important point to resolve as readers familiar with offline RL and reward-conditioned policies might be confused about the novelty or at least the relation to prior work when reading this paper.
>
> Thank you for your insightful comments. We acknowledge the significance of comparing our proposed method against other offline RL approaches to effectively demonstrate its effectiveness.
>
> We are thankful to you for the additional context you provided, particularly regarding the prior works on reward conditioning and reward shaping. In response, we have made necessary revisions to the paper to address these points. Specifically, we have updated the offline RL subsection of related works section to include a reference to the "RCP-A" paper, and we have also incorporated information about reward shaping in the Methodology section on page 6. We believe that these enhancements will address any potential confusion readers familiar with offline RL and reward-conditioned policies may have regarding the novelty and relation to prior work of our approach. Thank you for your invaluable suggestions, and we are grateful for the opportunity to improve the clarity and comprehensiveness of our paper.
>
> >I appreciate the new experiments, but I studied the baseline and I think it's more fair to initialize the returns-to-go token with something less than the maximum trajectory length at each timestep. By initializing it with the max, you are literally asking the model for poorer performance on at least half the trajectories assuming the max length isn't the same for all trajs. But in the decision transformer paper, they initialize test-time returns optimistically, not pessimistically, for best perofrmance. A more fair value could just be the average instead. Can the authors perform this simple comparison too?
>
> We agree that initializing the returns-to-go token with the maximum trajectory length may not always provide the most equitable comparison, especially if trajectory lengths vary across instances. Considering this, we had already conducted experiments with having average length as the test-time initial return. Specifically, the results for these experiments are included in Table 9, where we have provided a comparison between the two initialization strategies. The results illustrate that using the maximum length of trajectories as the initial return yields superior performance compared to using the average length. We apologize if this detail was not immediately apparent, and we appreciate the opportunity to clarify.
>
> Once again, we sincerely appreciate your valuable feedback and suggestions, which have undoubtedly improved the clarity and comprehensiveness of our paper. Thank you for your continued support and guidance.

---

> > ### Comment · Reviewer_5e26 · 2024-02-18
> >
> > I've taken a look at Table 9 and the other additions you just made, and I think that all of my concerns are addressed.
> >
> > I will be updating my recommendation to accept. Thanks.

---

### Decision · Action_Editor_YF8p · 2024-03-25

**Recommendation:** Accept as is

**Comment:**

The reviewers all agree that the paper is of interest to the TMLR audience and provides clear and convincing evidence for its claims.

One reviewer was left with a small concern: "the reward is designed to overfit the natural of RxR and not proven to be generalizable yet; adding noise to distillation is still kind of distillation; I still can not convince myself that HAMT resembles the actual noise in offline data".  The authors might want to consider this in their final revision.

**Audience:**

Yes.

**Claims And Evidence:**

Yes.